# Embedding Dimension of Contrastive Learning and $k$-Nearest Neighbors

**Dmitrii Avdiukhin**
Computer Science Department
Northwestern University
Evanston, IL 60657, USA
dmitrii.avdiukhin@northwestern.edu

**Vaggos Chatziafratis**
Computer Science and Engineering Department
University of California at Santa Cruz
Santa Cruz, CA 95064, USA
vaggos@ucsc.edu

**Orr Fischer**
Computer Science Department
Bar-Ilan University
Ramat-Gan, Israel
fischeo@biu.ac.il

**Grigory Yaroslavtsev**
Computer Science Department
George Mason University
Fairfax, VA 22030, USA
grigory@gmu.edu

## Abstract

We study the embedding dimension of distance comparison data in two settings: contrastive learning and $k$-nearest neighbors ($k$-NN). Our goal is to find the smallest dimension $d$ of an $\ell_p$-space in which a given dataset can be represented. We show that the *arboricity* of the associated graphs plays a key role in designing embeddings. For the most popular $\ell_2$-space, we get tight bounds in both settings.

In contrastive learning, we are given $m$ labeled samples $(x_i, y_i^+, z_i^-)$ representing the fact that the positive example $y_i$ is closer to the anchor $x_i$ than the negative example $z_i$ (we also give results for $t$ negatives). For representing such dataset in:

- $\ell_2$: $d = \Theta(\sqrt{m})$ is necessary and sufficient, consistent with our experiments.
- $\ell_p$ for $p \geq 1$: $d = O(m)$ is sufficient and $d = \tilde{\Omega}(\sqrt{m})$ is necessary.
- $\ell_\infty$: $d = O(m^{2/3})$ is sufficient and $d = \tilde{\Omega}(\sqrt{m})$ is necessary.

In $k$-NN, for each of the $n$ data points we are given an ordered set of the closest $k$ points. We show that for preserving the ordering of the $k$-NN for every point in:

- $\ell_2$: $d = \Theta(k)$ is necessary and sufficient.
- $\ell_p$ for $p \geq 1$: $d = \tilde{O}(k^2)$ is sufficient and $d = \tilde{\Omega}(k)$ is necessary.
- $\ell_\infty$ : $d = \tilde{\Omega}(k)$ is necessary.

Furthermore, if the goal is to not just preserve the ordering of the $k$-NN but also keep them as the nearest neighbors, then $d = \tilde{O}(\text{poly}(k))$ suffices in $\ell_p$ for $p \geq 1$.

## 1 Introduction

Embedding vectors play an important role in machine learning, with the embedding dimension being a key parameter of interest when choosing a deep learning architecture. In this paper, we ask the following question: given a dataset labeled with distance relationships between its points, what is the smallest embedding dimension required to represent it? We answer this question for two types of distance comparison data: contrastive labels and $k$-NN.

**Contrastive Learning**  Contrastive learning [GH10] has recently become a popular technique for learning representations, see e.g. [SE05, MCCD13, DSRB14, SKP15a, WG15, WXYL18,

38th Conference on Neural Information Processing Systems (NeurIPS 2024).

LL18, HFL$^+$19, HFW$^+$20, TKI20, CKNH20, CH21, GYC21, CLL21]. Recent interest in theoretical foundations of contrastive learning has resulted in extensive research focusing on generalization [AAE$^+$24], design of specific loss functions [HWGM21], transfer learning [SPA$^+$19, CRL$^+$20], multi-view redundancy [TKH21], inductive biases [SAG$^+$22, HM23], the role of negative samples [AGKM22, ADK22], mutual information [vdOLV18, HFL$^+$19, BHB19, TDR$^+$20], and other topics [WI20, TWSM21, ZSS$^+$21, vKSG$^+$21, MMW$^+$21, WL21].

In on of the most common forms of contrastive learning, we are given $m$ labeled data points $\{(x_i, y_i^+, z_i^-)\}_{i=1}^m$ (or more generally, $\{(x_i, y_i^+, z_{i,1}^-, z_{i,2}^-, \ldots, z_{i,t}^-)\}_{i=1}^m$) over a dataset of size $n$. Each point represents the fact that the distance between the *anchor* $x_i$ and the *positive example* $y_i$ is smaller than the distance between $x_i$ and the *negative example* $z_i$ (or, more generally, $t$ negative examples $z_{i,1}, \ldots, z_{i,t}$). We study the problem of embedding such data into $\ell_p$-spaces, i.e., constructing an embedding $F \colon V \to \mathbb{R}^d$ such that $\|F(x_i) - F(y_i)\|_p < \|F(x_i) - F(z_i)\|_p$ for all $i$ (more generally, $\|F(x_i) - F(y_i)\|_p < \|F(x_i) - F(z_{i,j})\|_p$ for all $i, j$). In particular, we focus on the embedding dimension:

*Given a collection of $m$ triplet comparisons of the form "$x_i$ is closer to $y_i$ than to $z_i$", what is the smallest dimension $d$ of an $\ell_p$-space in which the relative order of distances can be preserved?*

$k$-**NNs**  We also study a similar question for $k$-Nearest Neighbor ($k$-NN) data, which has major applications in machine learning since the seminal work of [CH67]. In this setting, we are given a set of $n$ items and the information about the $k$-NN of each item $\{(x_i, \pi_1(x_i), \ldots, \pi_k(x_i))\}_{i=1}^n$ where $\pi_1(x_i), \ldots, \pi_k(x_i)$ are the $k$-NN of $x_i$ ordered by their distance from $x_i$. Since $k$-NN classifiers are extremely popular in deep learning pipelines, understanding the embedding dimension required for preserving $k$-NN is a question of fundamental importance. In particular:

*Given $n$ items and their $k$-NN, what is the smallest dimension $d$ of an $\ell_p$-space in which the ordering of the $k$-NN can be preserved? What if the $k$-NN have to remain $k$-NN in the $\ell_p$-space?*

## 1.1   Our Results and Techniques

Let $V$ be the set of $n$ *points*. Our goal is to construct an embedding $F \colon V \to \mathbb{R}^d$. For an integer $n$, we let $[n] = \{1, 2, \ldots, n\}$. For a vector $v \in \mathbb{R}^d$, let $v[i]$ be the $i^{th}$ coordinate of $v$. For vectors $v_1, v_2$, we denote their concatenation as $(v_1, v_2)$. In a graph, denote by $N(x)$ the neighbors of vertex $x$. For standard definitions (e.g. *metric* and *norm*) and basic facts see Appendix B.

**Contrastive Learning**  For a set of samples $Q = \{(x_1, y_1^+, z_1^-), \ldots, (x_m, y_m^+, z_m^-)\}$, we call an embedding $F$ consistent with $Q$ if $\|F(x_i) - F(y_i)\|_p < \|F(x_i) - F(z_i)\|_p$ for all $i$. W.l.o.g., we can assume[1] that $m \le n^2$. We call a set of samples non-contradictory if one can't derive a contradiction from the inequalities between the distances. In particular, this implies the existence of a metric $\rho$ which is consistent with $Q$ (Fact 25).

We prove the following theorems in Section 2, Appendix D.2, and Appendix D respectively.

**Theorem 1** (Embedding in $\ell_2$). *Let $Q$ be a set of $m$ non-contradictory triplet samples on a set $V$. There is an embedding of $V$ into $\ell_2$-space $\mathbb{R}^{O(m^{1/2})}$ which is consistent with $Q$.*

**Theorem 2** (Embedding in $\ell_\infty$). *Let $Q$ be a set of $m$ non-contradictory triplet samples on a set $V$. There is an embedding of $V$ into $\ell_\infty$-space $\mathbb{R}^{O(m^{2/3})}$ which is consistent with $Q$.*

**Theorem 3** (Embedding in $\ell_p$). *Let $Q$ be a set of $m$ non-contradictory triplet samples on a set $V$. For any integer $p \ge 1$, there is an embedding of $V$ into $\ell_p$-space $\mathbb{R}^{O(m)}$ which is consistent with $Q$.*

The lower bounds are shown in Appendix E and experimental results are in Section 4. Our results for the more general version of the problem with $t$ negatives and the lower bounds are given in Table 1.

In Appendix F we give additional results, including an extension to $t$-negatives, NP-hardness of fining an embeddding in the minimum dimension needed to satisfy a set of contrastive constraints, and results for an approximate setting in which we only need to satisfy a fraction of the constraints.

---

[1]This is since $n^2$ triplet samples are enough to describe all comparisons – for each anchor, it suffices to know the order of other points w.r.t. their distance to the anchor. Hence, for any embeddable set of samples $Q$, there exists a set of at most $n^2$ samples which is also embeddable and at least as restrictive as $Q$.

Table 1: Our results for contrastive learning

| Setting | Upper bound | Lower bound |
|---|---|---|
| $\ell_2$ | $O(\sqrt{m})$, Theorem 1 | $\Omega(\sqrt{m})$, Theorem 43 |
| $\ell_2$ with $t$ negatives | $O(\sqrt{mt})$, Theorem 44 | $\Omega(\sqrt{m})$, Theorem 43 |
| $\ell_2$ with $t$-ordering | $O(\sqrt{mt})$, Theorem 44 | $\Omega(\sqrt{mt})$, Theorem 43 |
| $\ell_\infty$ | $O(m^{2/3})$, Theorem 2 | $\widetilde{\Omega}(\sqrt{m})$ Theorem 43 |
| $\ell_p$, integer $p \geq 1$ | $O(m)$, Theorem 3 | Even $p$: $\Omega(\sqrt{m})$, odd $p$: $\widetilde{\Omega}(\sqrt{m})$, Theorem 43 |

Table 2: Our results for $k$-NN

| Setting | Upper bound | Lower bound |
|---|---|---|
| $\ell_p$ ($k$-NN and ordering) | $\widetilde{O}(k^{10})$, Theorem 5 | even $p$: $\Omega(k)$, odd $p$: $\tilde{\Omega}(k)$, Theorem 43 |
| $\ell_p$ (ordering of $k$-NN) | $\widetilde{O}(k^2)$, Theorem 10 | |
| $\ell_2$ (ordering of $k$-NN) | $O(k)$ Theorem 6 | $\Omega(k)$[CI24] |
| $\ell_\infty$ (ordering of $k$-NN) | — | $\tilde{\Omega}(k)$, Theorem 43 |

$k$**-NN**    In the $k$-NN setting, we are given the following information for each data point.

**Definition 4** ($k$-NN). *For a distance function $\delta\colon V \times V \to \mathbb{R}_{\geq 0}$, let $\pi_1(x), \ldots, \pi_{n-1}(x)$ be an ordering of $V \setminus \{x\}$ such that $\delta(x, \pi_1(x)) < \delta(x, \pi_2(x)) < \cdots < \delta(x, \pi_{n-1}(x))$. We define $\text{k-NN}_\delta(x) = (\pi_1(x), \ldots, \pi_k(x))$ as the ordered set of $k$ closest points to $x$.*

For a function $F\colon V \to \mathbb{R}^d$, we denote by $\text{k-NN}_F$ the $k$-nearest neighbors in the $\ell_p$-space corresponding to the image of $F$. We prove the following theorem in Section 3.

**Theorem 5.** *Let $\delta\colon V \times V \to \mathbb{R}_{\geq 0}$ be a distance function, and let $p \geq 1$ be a constant. There exists an embedding $F\colon V \to \mathbb{R}^d$ of $V$ into an $\ell_p$-space of dimension $d = O(k^{10} \log^{10} n)$ such that $\text{k-NN}_\delta(x) = \text{k-NN}_F(x)$, i.e. the embedding $F$ preserves the ordered set of $k$-nearest neighbors of any point $x \in V$ under the distance function $\delta$.[2]*

We note that the above result is very surprising: $k$-NN graph in fact corresponds to $n(n-1)$ triplet constraints – for each anchor, $k-1$ comparisons between its $k$-NN and $n-k$ comparisons between the $k$'th nearest neighbor and the rest of the points – and Theorem 1 provides only an $O(n)$ upper bound on dimension for the $\ell_2$ case. Nevertheless, we are able to exploit the structure of the contrastive constraints to avoid polynomial dependence on $n$.

The following theorem addresses the setting when only the ordering of the $k$-NN has to be preserved. This, as well as other results for $k$-NN, are presented in Table 2.

**Theorem 6.** *There is an embedding of $V$ into $\ell_2$-space $\mathbb{R}^{O(k)}$ that preserves the $k$-NN ordering.*

**Our Techniques**    The key tool in our results is the notion of graph *arboricity* [NW61, NW64] applied to the associated *constraint graph*. Arboricity of an undirected graph is the minimum number of forests in which its edges can be partitioned. More intuitively, arboricity measures the "density" of the graph: sparse graphs have low arboricity, while graphs with dense subgraphs – such as cliques – have high arboricity.

**Fact 7** (Folklore; see e.g. [BE13, DHS91] and Appendix B.2). *The arboricity $r$ of a graph $G$ with $m$ edges is at most $\lceil \sqrt{m}/2 \rceil$. Moreover, if graph $G$ has arboricity $r$, then the following hold.*

*(a) There is an ordering $x_1, \ldots, x_n$ of $V$ such that $|N^-(x_i)| \leq 2r - 1$ for each $1 \leq i \leq n$, where $N^-(x_i) = \{x_j \in N(x_i) \mid j < i\}$ is the set of neighbors of $x_i$ in $G$ preceding $x_i$ in the ordering.*

*(b) $G$ is $2r$-vertex colorable.*

**Definition 8** (Constraint graph). *In contrastive learning, for a set $Q$ of samples on $V$, we define the* constraint graph *$G = (V, E)$ as follows: for each sample $(x_i, y_i^+, z_i^-) \in Q$, we add two edges $\{x_i, y_j\}$ and $\{x_i, z_i\}$ to $E$, unless they already exist. In the $k$-NN setting, for each $x$ and its nearest neighbors $\pi_1(x), \ldots, \pi_k(x)$, we add edges $\{x, \pi_i(x)\}$ for $1 \leq i \leq k$.*

---

[2]In subsequent versions of our paper, we have improved the analysis to show a dimension bound of $\widetilde{O}(k^3)$.

Note that by Fact 7 the arboricity of the constraint graph resulting from $m$ samples is at most $\sqrt{m}$. The arboricity of the $k$-NN constraint graph is at most $k + 1$ (See Lemma 27). We show bounds on the embedding dimension in terms of arboricity, e.g. for $\ell_2$ we prove the following in Section 2.

**Theorem 9.** *Given a set of non-contradictory inequalities among pairwise distances in $V$ whose constraint graph has arboricity $r$, there exists an embedding of $V$ into $\ell_2$-space $\mathbb{R}^{4r}$ which satisfies all these inequalities.*

Theorem 1 follows from Theorem 9 by using $r \leq \lceil \sqrt{m}/2 \rceil$ (Fact 7). Moreover, since the arboricity of the constraint graph for k-NN at most $k + 1$ (Lemma 27), Theorem 9 shows that preserving the ordering of the $k$-NN in $\ell_2$ requires $O(k)$ dimension. Furthermore, the following theorem, proven in Section 3.1, implies that $\tilde{O}(k^2)$ dimension suffices to preserve orderings of the k-NNs in $\ell_p$.

**Theorem 10.** *Given a set of non-contradictory inequalities among pairwise distances in $V$ whose constraint graph has arboricity $r$, for any real $p \geq 1$, there exists an embedding of $V$ into $\ell_p$-space $\mathbb{R}^{O(r^2 \log^3 n)}$ which satisfies all these inequalities.*

While the above constructions suffice for the contrastive learning case and for preserving the *ordering* of the $k$-NN, the *set* of the nearest neighbors can change under the embeddings above. Hence, in order to preserve the $k$-NN, we increase the dimension to separate neighbors from non-neighbors. In particular, we construct the extended part of the embedding randomly, using a sampling scheme which is guaranteed to embed neighbors much closer than non-neighbors. See Section 3.2 for more details and a proof of Theorem 5.

For $\ell_\infty$, instead of arboricity, we use a related fact: by removing a set $V_{\text{high}}$ of $O(m^{2/3})$ high-degree vertices, we reduce the maximum degree of the remaining graph (i.e. $V_{\text{low}} = V \setminus V_{\text{high}}$) to at most $O(m^{1/3})$. We handle each set differently (points in $V_{\text{low}}$ using graph colorings, and points in $V_{\text{high}}$ using a Frechét-like embedding). See Appendix D.2 for the details and the proof of Theorem 2.

## 1.2 Previous Work

Understanding the underlying geometry of a given set of $n$ points based only on comparisons between pairs of distances is a basic question studied in the literature of non-metric embeddings (also known as ordinal embeddings or monotone maps). In a wide range of applications such as ranking, crowdsourcing, nearest-neighbor search, ad placement, recommendation systems, etc., the exact distances are not as important as their relative order. In fact, some of the early results in the field were motivated by applications in mathematical psychology [Tor52, She62, She74, CS74, Kru64a, Kru64b], and since then ordinal information and embeddings have been used in ranking [OG08, Ail12, WJJ13], metric learning [CHX+19], clustering [VD16, GPvL19, KVH16], crowdsourcing [TLB+11, JN11a, JN11b] and modeling human perception [ML09]. Note that the goal in ordinal embeddings is quite different from the vast literature on metric embeddings (e.g., see [Mat13, IMS17]) where the goal is to approximately preserve the numerical values of distances.

We study the question of finding the smallest dimension $d$ required to represent a given set of $n$ points such that a given set of $m$ distance comparisons are preserved. Related questions have been studied under statistical assumptions and it is known [KL14, TL14, GCY19] that for the large $n$ regime, upon knowledge of the ordinal relationships, the set of points can be approximately recovered (up to certain transformations). This serves as further motivation for studying ordinal information as it highlights its power in recovering the underlying geometry of the data points.

However, determining the exact relationships between the dimension $d$, the number of points $n$ and the number of given constraints $m$ has been elusive. Most papers assume that all $\Theta(n^4)$ distance comparisons $\delta(x_i, x_j) \lesseqgtr \delta(x_k, x_l)$ among the pairwise distances are known. In [BL05, ABD+08, BDH+08], for example, lower bounds are given for the dimension needed to preserve these comparisons. However, having access to such a large number of comparisons is prohibitive in practice. We only assume access to a set of $m$ distance comparisons and hence these lower bounds do not apply.

Contrastive learning has been studied for $d = 1$ (embedding on the line) by [FIM+20] for dense instances, i.e. $m = \Theta(n^3)$. For higher dimensions, [CI24] gives an $\Omega(n)$ lower bound on the smallest dimension (only for $\ell_2$) that preserves all $\Theta(n^3)$ triplet comparisons. Our Theorem 1 improves this bound for the general case when $m$ triplet samples are given, without density assumptions. Then,

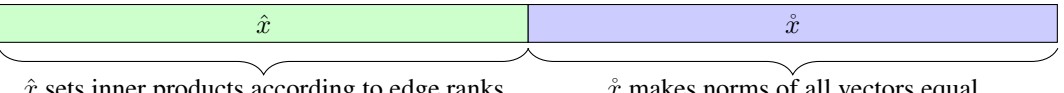

$\hat{x}$ sets inner products according to edge ranks    $\mathring{x}$ makes norms of all vectors equal

Figure 1: $\hat{x}$ is chosen so that if $w(x,y) > w(x',y')$, then $\langle \hat{x}, \hat{y} \rangle < \langle \hat{x}', \hat{y}' \rangle$. $\mathring{x}$ ensures that all vectors have the same norm, i.e. $\|\mathring{x}\|_2^2 = W$ for all $x \in V$.

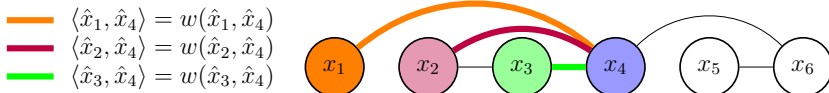

Figure 2: Example construction of $\hat{x}$. The embedding $\hat{x}_4$ is computed based on the embeddings of its already processed neighbors $\hat{x}_1, \hat{x}_2, \hat{x}_3$. We find the solution $\hat{x}_4$ to the linear system so that, for each edge to a preceding neighbor, the inner product equals the rank of the edge.

our Theorems 2 and 3 go beyond $\ell_2$ other $\ell_p$-norms. Our results can also be seen as the reverse direction of the recent work by [AAE+24]. In [AAE+24], the central question is quantifying the amount of data required for generalization in contrastive learning, assuming that the data can be embedded into an $\ell_p$-space of fixed dimension. Here we assume that the data is fixed instead and study the embedding dimension. Combined with [AAE+24], this completes the picture of the relationship between the size of data, its embedding dimension and generalization.

Our second setting ($k$-NNs) was also studied in [CI24] who showed a lower bound of $d = \Omega(k)$ for preserving the ordering of the neighbors (again in $\ell_2$). To the best of our knowledge, prior to our work, there was no known upper bound for the smallest dimension and here we provide a matching upper bound. Furthermore, we provide new results for $k$-NNs embeddings (both upper and lower bounds) under various $\ell_p$ metrics and results for the stronger setting when not just the ordering of the neighbors but also their status as $k$-NN has to be preserved.

## 2 Contrastive Learning in $\ell_2$ Norm

In this section, we prove Theorem 9 – that contrastive queries with the constraint graph $G = (V, E)$ (Definition 8) of arboricity $r$ are preserved when the points are embedded into $\ell_2$ space of dimension $4r$ – from which Theorem 1 and Theorem 6 follow. Fix a distance function $\delta \colon V \times V \to \mathbb{R}_{\geq 0}$ that satisfies the given set of inequalities (such a function exists by Fact 25). We order all pairs of neighboring vertices by the distance function $\delta$ in descending order, and let $w(x,y) = i$ if $\{x, y\}$ is the $i$-th pair in the ranking. Recall that $\|F(x) - F(y)\|^2 = \|F(x)\|^2 + \|F(y)\|^2 - 2\langle F(x), F(y) \rangle$. In our construction, all embeddings have the same norm, and hence the distances depend only on the inner products between the embeddings.

We split the embedding $F \colon V \to \mathbb{R}^{4r}$ into two parts, i.e. for a point $x$ let $F(x) = (\hat{x}, \mathring{x})$, where $\hat{x} \in \mathbb{R}^{2r}$ and $\mathring{x} \in \mathbb{R}^{2r}$. For neighboring points $x$ and $y$, our choices of $\hat{x}$ and $\hat{y}$ ensure that $\langle \hat{x}, \hat{y} \rangle \approx w(x,y)$. We embed the points one by one into $\mathbb{R}^h$ in the arboricity ordering $x_1, \ldots, x_n$, which by Fact 7 ensures that for every vertex, the number of neighbors with smaller indices is at most $h$. When embedding $x_i$, we make sure that for any neighbor $x_j \in N^-(x_i)$ (i.e. a neighbor $x_j$ of $x_i$ such that $j < i$) it holds that $\langle \hat{x}_i, \hat{x}_j \rangle \approx w(x_i, x_j)$. This requires solving a linear system over $\hat{x}_i$ with at most $h$ equations, and hence with $h$ variables, with slight perturbations, the solution always exists.

The choices of $\mathring{x}_i$ ensure that all vectors have the same norm while preserving the inner products. This is done by coloring the vertices of the constraint graph in $h$ colors using Fact 7 and assigning each color to a unique basis vector, which is scaled to equalize the norms. Since these basis vectors are orthogonal, the inner product between any two neighboring points $x_i$ and $x_j$ is $\langle \hat{x}_i, \hat{x}_j \rangle$.

**Construction of** $\hat{x}_i$   Assume $\hat{x}_1, \ldots, \hat{x}_{i-1}$ have already been chosen. Let $N^-(x_i) = \{x_j \in N(x_i) \mid j < i\}$ be the set of preceding neighbors of $x_i$ in $G$. For each $x_j \in N^-(x_i)$, let a linear equation $P(i, j)$ be $\langle \hat{x}_i, \hat{x}_j \rangle = w(x_i, x_j)$, where we consider the coordinates of $\hat{x}_i$ as variables (recall that $\hat{x}_j$ is already set for all $x_j \in N^-(x_i)$). In Appendix C we show the following.

**Lemma 11.** *The set of vectors $\{\hat{x}_j \mid x_j \in N^-(x_i)\}$ is linearly independent.*

By Lemma 11, the system of linear equations $P_i = \{P(i, j) \mid x_j \in N^-(x_i)\}$ has a solution $v \in \mathbb{R}^{2r}$. Let $B(v)$ be a ball centered at $v$ with sufficiently small radius such that for any $v' \in B(v)$ it holds that $|\langle v', \hat{x}_j \rangle - w(x_i, x_j)| < 1/3$ for all $x_j \in N^-(x_i)$. Choose a point $v'$ uniformly at random from $B(v)$, and set $\hat{x}_i = v'$: this random perturbation guarantees that, with probability 1, Lemma 11 holds in future iterations. By construction, the following property holds.

**Proposition 12.** *For any $x$ and any $y \in N^-(x)$, we have $|\langle \hat{x}, \hat{y} \rangle - w(x, y)| < 1/3$.*

**Construction of $\mathring{x}_i$** Let $W = 2\max_{x \in V} \|\hat{x}\|_2^2$. By Fact 7, there exists vertex coloring $C: V \to [h]$ of $G$, such that $C(x) \neq C(y)$ for any pair $\{x, y\} \in E$. Set $\mathring{x} = \alpha_x e_{C(x)}$, where $e_{C(x)}$ is the standard basis vector in the $C(x)$-th coordinate, and $\alpha_x$ is chosen so that $\|F(x)\|_2^2 = \|\hat{x}\|_2^2 + \|\mathring{x}\|_2^2 = W$ (note that $\alpha_x$ exists because $\|\hat{x}\|_2^2 \leq W$). By construction, the following property holds.

**Proposition 13.** *For any edge $\{x, y\} \in E$, we have $\langle \mathring{x}, \mathring{y} \rangle = 0$.*

*Proof of Theorem 9 (sufficient dimension for $\ell_2$ embeddings).* For any edge $\{u, v\} \in E$ it holds that

$$\|F(u) - F(v)\|_2^2 = \|F(u)\|_2^2 + \|F(v)\|_2^2 - 2\langle \hat{u}, \hat{v} \rangle - 2\langle \mathring{u}, \mathring{v} \rangle.$$

By the choice of $\mathring{u}$ and $\mathring{v}$, we have $\|F(u)\|_2^2 = \|F(v)\|_2^2 = W$. By Proposition 13, $\langle \mathring{u}, \mathring{v} \rangle = 0$, and hence the distance depends only on $\langle \hat{u}, \hat{v} \rangle$. For any $(x, y^+, z^-) \in Q$, we have $\|F(x) - F(y)\|_2^2 < \|F(x) - F(z)\|_2^2$ iff $\langle \hat{x}, \hat{y} \rangle > \langle \hat{x}, \hat{z} \rangle$. By Proposition 12, for any edge $\{x, y\}$ in $G$ it holds that $|\langle \hat{x}, \hat{y} \rangle - w(x, y)| < 1/3$. Since the function $w$ assigns only integer values, it holds that $\langle \hat{x}, \hat{y} \rangle > \langle \hat{x}, \hat{z} \rangle$ if and only if $w(x, y) < w(x, z)$, hence preserving the ranking of the edges. $\square$

## 3 Preserving $k$ Nearest Neighbors

In this section, we focus on $k$ nearest neighbors, and namely we prove Theorems 5 and 10. Let $G = (V, E)$ be the constraint graph (Definition 8) for given $k$-NN input. In Section 3.1, we show how to preserve the order between the neighbors in this graph, and in Section 3.2 we show how to separate neighbors from non-neighbors. Combined, these results fully preserve the $k$-NNs.

To simplify the presentation, we focus on the case $p = 1$ – the construction for other $p$ is identical, with the change being that each embedding coordinate value $c$ should be replaced with $c^{1/p}$. In this section, let $\delta(u, v) = \delta_{\ell_1}(u, v)$. For a non-contradictory set of samples $Q$, by Fact 25 there exists a metric $\delta'$ consistent with $Q$. We order all pairs of neighboring vertices by the value of $\delta'$ in descending order, and let $w(x, y) = t$ if $(x, y)$ is the $t$-th pair in the ranking. Given an embedding $F$, let $\alpha F$ be a re-scaling of the embedding by a factor of $\alpha$, i.e. multiplying each coordinate by $\alpha$.

### 3.1 Preserving the Ordering of the k-NN

In this section, we show Theorem 10. This embedding is also used as a part of Theorem 5, shown in Section 3.2. Our embedding uses a new coloring scheme we call *Neighbor-Collection Coloring*. Let $x_1, \ldots, x_n$ be the arboricity ordering (Fact 7) and $N^-(x_i) = \{x_j \mid \{x_i, x_j\} \in E, j < i\}$ be the set of neighbors of $x_i$ preceding $x_i$ in the ordering.

**Definition 14** (NCC Scheme). *A neighbor-collection coloring scheme is a set of $K = \Theta(r \log n)$ vertex colorings $C^{(1)}, \ldots, C^{(K)}$, where $C_x^{(j)} \in [r]$ for any $x \in V$ and $j \in [K]$, such that for any $x \in V$ the following holds:*

- *(Collection) for any $y \in N^-(x)$, there exists a coloring $j \in [K]$ such that $C_x^{(j)} = C_y^{(j)}$, and $C_z^{(j)} \neq C_x^{(j)}$ for any $z \in N^-(x) \setminus \{y\}$.*

- *(Load) for any $j \in [K]$, the number of prior neighbors with $j$-th color being the same as $C_x^{(j)}$ is small: $|\{y \in N^-(x) \mid C_x^{(j)} = C_y^{(j)}\}| = O(\log n)$.*

Intuitively, each coloring corresponds to a part of the embedding. When the colors $C_x^{(j)}, C_y^{(j)}$ are different, the $j$'th part of the embedding always contributes 2 to the distance between $x$ and $y$. Otherwise, we can select the $j$'th part so that it contributes either 2 or 0, and the collection property

guarantees that for any $y \in N^-(x)$ such a part exists. The load property guarantees that for each part we always have enough choices to get distance 2. Finally, we represent $w(x, y)$ in binary format for all $x, y$, and, using an NCC scheme, we recover $w(x, y)$ bit-by-bit.

**Lemma 15.** *There exists an NCC scheme for the constraint graph $G$.*

*Proof.* For each $x \in V$ and $j \in [K]$, we choose $C_x^{(j)}$ i.i.d. uniformly at random from $[r]$. First, note that the load property holds: for any $j \in [K]$ and $y \in N^-(x)$, we have $\mathbb{P}\left[C_x^{(j)} = C_y^{(j)}\right] = 1/r$. By Fact 7, we have $|N^-(x)| \leq 2r$, and by the Chernoff bound, color $C_x^{(j)}$ occurs no more than $O(\log n)$ times in $N^-(x)$ w.h.p. By the union bound, the load property holds w.h.p. for all $j$.

Next, for any fixed $x \in V$, $y \in N^-(x)$, and $j \in [K]$, let $A^{(j)}(x, y)$ be the event that $y$ is the only point in $N^-(x)$ such that $C_x^{(j)} = C_y^{(j)}$. Since the colorings are selected uniformly at random, we have $\mathbb{P}\left[A^{(j)}(x, y)\right] = \Omega(1/r)$. Since $K = O(r \log n)$, by Chernoff, w.h.p. there exists $j \in [K]$ such that $A^{(j)}(x, y)$ occurs. By the union bound, the collection property holds w.h.p. $\quad\square$

**Definition 16** (NCC-Embedding). *Given a graph $G$ and an NCC scheme, an NCC-embedding is an embedding of dimension $O(r^2 \log^2 n)$ of the following form. Associate each color $i \in [r]$ with $M = O(\log n)$ unique basis vectors $\mathcal{B}(i) = \{e_{(i-1)M+1}, e_{(i-1)M+2}, \ldots, e_{iM}\}$. The embedding of point $x$ is comprised of $K$ parts $\mathring{x}^{(1)}, \ldots, \mathring{x}^{(K)}$, where each part is a basis vector $\mathring{x}^{(j)} \in \mathcal{B}(C_x^{(j)})$, i.e. $\mathring{x}^{(j)}$ is one of the basis vectors associated with color $C_x^{(j)}$.*

**Lemma 17.** *Let $D \colon E \to \{0, 1\}$ be a mapping of each edge, with $1$ meaning "close" and $0$ meaning "far". For each $x \in V$ there exists embedding $\mathring{x}$ into $O(r^2 \log^2 n)$ dimensions such that for any $\{x, y\} \in E$, it holds that $\delta(\mathring{x}, \mathring{y}) = K - D(x, y)$.*

*Proof.* Let $(C^{(1)}, \ldots, C^{(K)})$ be an NCC scheme of $G$. We embed the points one by one according to the arboricity ordering $x_1, \ldots, x_n$ as in Fact 7. We assume by induction that all nodes $x_1, \ldots, x_{i-1}$ are embedded using an NCC-embedding. For each $y \in N^-(x)$, fix one index $j(y)$ such that under $C^{(j(y))}$ the points $x, y$ have the same color, which is different from colors of other points from $N^-(x)$ (such $j(y)$ exists by the collection property). Let $J = \{j(y) \mid y \in N^-(x)\}$, and, since for any two points in $N^-(x)$ the chosen index is distinct, $|J| = |N^-(x)|$.

For each part $j \in [K] \setminus [J]$, we choose $x^{(j)}$ to be a basis vector from $\mathcal{B}(C_x^{(j)})$ that is different from all basis vectors $\{\mathring{y}^{(j)} \mid y \in N^-(x)\}$. This can be done, since, on the one hand, for each $C_x^{(j)} \neq C_y^{(j)}$, all basis vectors of $\mathcal{B}(C_x^{(j)})$ are different from $\mathring{y}^{(j)}$, and, on the other hand, by the load property there are less than $O(\log n)$ points $y \in N^-(x)$ such that $C_x^{(j)} = C_y^{(j)}$. Therefore, we can choose a basis vector that is different from any taken by these $O(\log n)$ points.

For each part $j(y) \in [J]$, we select the basis vector based on $D$. If $D(x, y) = 1$, then we take $\mathring{x}^{(j(y))} = \mathring{y}^{(j(y))}$. Otherwise, we pick a basis vector $\mathring{x}^{(j(y))} \in \mathcal{B}(C_x^{(j(y))})$ such that $\mathring{x}^{(j(y))} \neq \mathring{y}^{(j(y))}$.

We now show that distance between embeddings is $2(K-1)$ if the points are close, and is $2K$ otherwise. The result follows by scaling the embedding. Let $\{x, y\} \in E$ such that $y \in N^-(x)$. Let $I^{(j)}(x, y) = 1$ if $\mathring{x}^{(j)} \neq \mathring{y}^{(j)}$, and $I^{(j)}(x, y) = 0$ otherwise. Since each part is a basis vector, $\delta(\mathring{x}, \mathring{y}) = 2 \sum_{j \in [K]} I^{(j)}(x, y)$. By construction, for any $j \in [J] \setminus \{j(y)\}$ it holds that $I^{(j)}(x, y) = 1$. For $j(y)$ we have $I^{(j(y))}(x, y) = 1 - D(x, y)$, i.e. $\delta(\mathring{x}, \mathring{y}) = 2(K - D(x, y))$. Rescaling the embedding vectors by a factor of $1/2$ completes the proof. $\quad\square$

**Corollary 18.** *Let $a'$ be a power of $2$ such that for all $\{x, y\} \in E$ we have $a_{x,y} \in \{0, \ldots, a'\}$. Then there exists an embedding $F$ of $V$ into $O(r^2 \log^2 n \log a')$ dimensions such that for any $\{x, y\} \in E$, we have $\delta(F(x), F(y)) = K(a' - 1) - a_{x,y}$.*

*Proof.* Let $\text{Bin}^{(i)}(x, y)$ be the $i$'th bit of the binary encoding of $a_{x,y}$ using a string of size $\log_2 a'$ bits. Let $F_1, \ldots, F_{\log_2 a'}$ be embeddings as in Lemma 17, where for each $F_i$ we choose $D_i =$

$\text{Bin}^{(i)}(x, y)$. For embedding $F(x) = (F_1(x), 2F_2(x), \ldots, 2^i F_i(x), \ldots, (a'/2)F_{\log_2 a'}(x))$ we have

$$\delta(F(x), F(y)) = \sum_{i=1}^{\log_2 a'} \left( K - \text{Bin}^{(i)}(x, y) \right) \cdot 2^{i-1}$$

$$= K \sum_{i=1}^{\log_2 a'} 2^{i-1} - \sum_{i=1}^{\log_2 a'} \text{Bin}^{(i)}(x, y) \cdot 2^{i-1} = K(a' - 1) - a_{x,y}. \qquad \square$$

Theorem 10 follows immediately from Corollary 18 by taking $a_{x,y} = w(x, y)$ and $a' \geq m'$.

## 3.2 Fully Preserving k-NN

In this section, we prove Theorem 5, which states the existence of an embedding with dimension $d = O(k^{10} \log^{10} n)$ that preserves the $k$-NN. Our approach can be summarized as follows: for each $x \in V$, the final embedding is $F(x) = (2m\hat{x}, \mathring{x})$ (Figure 3). The goal of $\hat{x}$ is to have all non-neighbors $\{x', y'\} \notin E$ be at a larger distance than any neighbors $\{x, y\} \in E$, i.e. for some large $W$ it holds that $\delta(\hat{x}, \hat{y}) + W < \delta(\hat{x}', \hat{y}')$. The goal of $\mathring{x}$ is to order the distances of neighboring pairs $\{x, y\} \in E$ according to their rank, while still keeping non-neighbors further away than neighbors.

We choose $\hat{x}^{(j)}$ via a random process, so that for any two neighbors $\{x, y\} \in E$ we have $\hat{x}^{(j)} = \hat{y}^{(j)}$ with some probability $p_1$ (and otherwise they have substantial distance), while for non-neighbors $\{x, y\} \notin E$, we have $\hat{x}^{(j)} = \hat{y}^{(j)}$ with much smaller probability $p_2 \ll p_1$. Repeating this process, we get a separation in distances between neighbors and non-neighbors.

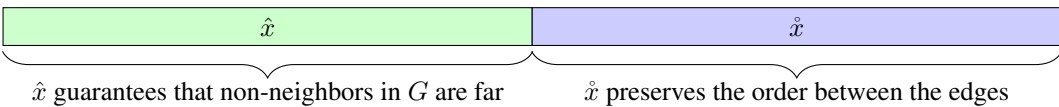

$\hat{x}$ guarantees that non-neighbors in $G$ are far   $\mathring{x}$ preserves the order between the edges

Figure 3: Structure of embedding for fully preserving k-NN. $\hat{x}$ guarantees that non-edges have very large distance, i.e. if $\{x, y\} \in E$ and $\{x', y'\} \notin E$, then $\delta(\hat{x}, \hat{y}) \ll \delta(\hat{x}', \hat{y}')$. $\mathring{x}$ orders the edges.

**Choosing $\hat{x}$:**  The embedding $\hat{x}$ is comprised of $L = \Theta(r^4 \log^4 n)$ parts, i.e. $\hat{x} = (\hat{x}^{(1)}, \ldots, \hat{x}^{(L)})$. We take each part $\hat{x}^{(j)}$ to be a vector from a *design* [DKS12] – a large family of vectors which are approximately equidistant.

**Definition 19** (($\alpha, R$)-design). *For integer $R$ and value $0 < \alpha < 1$, an $(\alpha, R)$-design is a family of sets $\mathcal{T}$, such that (a) for each $S_i \in \mathcal{T}$, $S_i \subseteq [R^2]$, (b) for each $S_i \in \mathcal{T}$, $|S_i| = R$, and (c) for each two distinct sets $S_i, S_j \in \mathcal{T}$, $|S_i \cap S_j| \leq \alpha R$.*

**Lemma 20** (Lemma 1, [DKS12]). *For any sufficiently large integer $R$ and any value $0 < \alpha < 1$, there exists an $(\alpha, R)$-design $\mathcal{T}$ of size at least $2^{\alpha R \log_2 R}$.*

Let $\mathcal{T}$ be a $(\alpha, R)$-design for $R = \Theta(r^3 \log^3 n)$ and $\alpha = \Theta(\log n / R)$, where $r$ is arboricity of the constraint graph (constants specified below). We associate $S \in \mathcal{T}$ with a binary vector $I(S) \in \{0, 1\}^{R^2}$ as an indicator vector of the set $S$, i.e. for $i \in [R^2]$ we have $I(S)[i] = 1$ iff $i \in S$. For each $x \in V$, we choose unique sets $S_x, S'_x \in \mathcal{T}$ and denote $I_x = I(S_x), I'_x = I(S'_x)$. By Lemma 20, the number of sets is $2^{\alpha R \log_2 R} = 2^{\Omega(\log n)}$, exceeding $2n$ for appropriate choices of constants.

We choose each part $\hat{x}^{(j)}$ independently of the rest as follows. For $p = O(1/(r \log n))$, with probability $1 - p$, we choose $\hat{x}^{(j)} = I_x$, and otherwise choose a uniformly random $i \in [2r]$. If $i \leq |N^-(x)|$, set $\hat{x}^{(j)} = I_y$, where $y \in N^-(x)$ is the $i$'th point in $N^-(x)$ according to some ordering, and set $\hat{x}^{(j)} = I'_x$ otherwise. Let $\gamma = \frac{(1-p)p}{2r}$ be the probability that $x$ and $y$ choose $I_y$.

Importantly, in this construction, neighbors are significantly more likely to sample the same vector compared with non-neighbors. Moreover, sampling the same vector contributes $0$ to the distance between embedding, while sampling different vectors contributes at least $(2 - \alpha)R$ to the distance. For $K$ is defined as in Definition 14, let $c = \max(\frac{r \log n}{100K}, \frac{1}{100})$ be a constant, and set $\alpha \leq \frac{c\gamma}{8r \log_2 n}$ and $R = \lceil \log_2 n / \alpha \rceil$. In Appendix C.1 we justify these choices of parameters and show the following.

**Lemma 21.** *With high probability, the following bounds hold.*

- *If $\{x, y\} \notin E$, then $|\delta(\hat{x}, \hat{y}) - 2RL| \leq \frac{c}{r \log_2 n} \gamma RL$*

- *If $\{x, y\} \in E$, then $|\delta(\hat{x}, \hat{y}) - 2(1 - \gamma)RL| \leq \frac{c}{r \log_2 n} \gamma RL$.*

*That is, according to the embedding, the gap between neighbors' distances and non-neighbor' distances is larger than the maximum difference between neighbors' distances.*

The final dimension is $O(r^{10} \log^{10} n)$: $L = \Theta(r^4 \log^4 n)$ parts of dimension $R^2 = \Theta(r^6 \log^6 n)$. Since $r = O(k)$ (Lemma 27), it follows that the dimension is bounded by $O(k^{10} \log^{10} n)$.

**Final Embedding**    Let $\mathring{x}_1, \ldots, \mathring{x}_n$ be the embeddings from Corollary 18 with $a'$ being the closest power of two from above of the expression $\frac{5mc\gamma}{r \log n} RL$. These embeddings have dimension at most $O(r^2 \log^2 n \log a') = O(r^2 \log^3 n)$. For $\{x, y\} \in E$, let $\Delta(x, y) = \left\lceil 2m \left(\delta(\hat{x}, \hat{y}) - 2\left(1 - \gamma - \frac{\gamma}{100}\right) RL\right) \right\rceil$. Set $a_{x,y} = \Delta(x, y) + w(x, y)$, where $w(x, y)$ is the ranking of edge $\{x, y\}$ if the edges are sorted by the decreasing order of distances. By Lemma 21, we have $0 \leq \Delta(x, y) \leq \frac{4mc\gamma}{r \log n} RL$ w.h.p., and hence $a_{x,y} \leq \frac{4mc\gamma}{r \log n} RL + m \leq a'$. Finally, $F(x) = (2m\hat{x}, \mathring{x})$.

*Proof of Theorem 5.* For each $x \in V$, let $F(x) = (2m\hat{x}, \mathring{x})$. It suffices to show the following.

(a) For any $\{x, y\} \in E$ and $\{x', y'\} \in E$, it holds that $w(x, y) < w(x', y')$ if and only if $\delta(F(x), F(y)) > \delta(F(x'), F(y'))$.

(b) For any $\{x, y\} \in E$ and $\{x', y'\} \notin E$, it holds that $\delta(F(x), F(y)) < \delta(F(x'), F(y'))$.

By Corollary 18, for any $\{x, y\} \in E$:

$$\delta(F(x), F(y)) = K(a' - 1) - \left\lceil 2m\left(\delta(\hat{x}, \hat{y}) - 2\left(1 - \gamma - \frac{\gamma}{100}\right) RL\right) \right\rceil - w(x, y) + 2m\delta(\hat{x}, \hat{y})$$

$$= K(a' - 1) + 4m\left(1 - \gamma - \frac{\gamma}{100}\right) RL - w(x, y) - \varepsilon_{x,y}, \tag{1}$$

where $\varepsilon_{x,y} \in [0, 1)$ is the rounding error. Hence, property (a) holds: if $w(x, y) < w(x', y')$ then $\delta(F(x), F(y)) > \delta(F(x'), F(y'))$, and vice versa, since the comparison is defined by ranking. The property (b) holds since for any $\{x', y'\} \notin E$ and $\{x, y\} \in E$:

$$\delta(F(x'), F(y')) \geq \delta(2m\hat{x}', 2m\hat{y}') \geq 4m\left(1 - \frac{\gamma}{100}\right) RL$$

$$\geq K(a' - 1) + 4m\left(1 - \gamma - \frac{\gamma}{100}\right) RL > \delta(F(x), F(y)),$$

where the second inequality follows from Lemma 21, and the third inequality follows from $K(a' - 1) \leq 4\gamma m RL$, which holds: since $a' - 1 \leq \frac{10c}{r \log n} \gamma m RL$, it suffices to have $K \leq \frac{4}{10c} r \log n$, which indeed holds for our choice of $c = \max(\frac{r \log n}{100 K}, \frac{1}{100})$. $\qquad\square$

## 4   Experiments

We perform experiments on CIFAR-10 and CIFAR-100 image datasets [KH09] (we show additional experiments in Appendix A). We define the ground-truth distance between points as the distance between their embedding vectors produced by a pretrained ResNet-18 neural network. Let $Q$ be contrastive triplets sampled uniformly at random from all possible triplets of images, labeled based on the ground-truth distance. Then, we train a different ResNet-18 model from scratch, where we control the embedding dimension by replacing the last fully-connected layer with a fully-connected layer with the chosen output dimension. We train the model for 50 epochs on a single NVIDIA A100 GPU using triplet loss [SKP15b]: $\mathcal{L}_F(x, y^+, z^-) = \|F(x) - F(y)\|^2 - \|F(x) - F(z)\|^2 + 1$. Since our goal is to find an embedding of this set of queries, we evaluate the accuracy as the fraction of satisfied contrastive samples.

We present our results in Figure 4. In experiments, we vary the number of samples (Figures 4a and 4b) and the dimension (Figures 4c and 4d). Figures 4a and 4b show that, while $d \geq \sqrt{m}$,

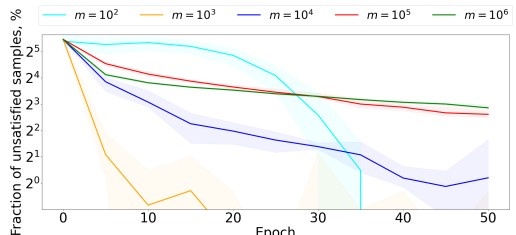
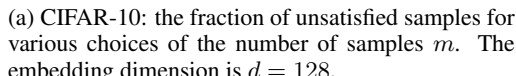
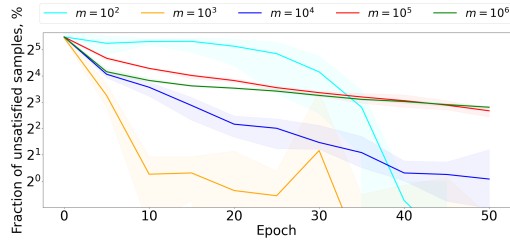

(a) CIFAR-10: the fraction of unsatisfied samples for various choices of the number of samples $m$. The embedding dimension is $d = 128$.

(b) CIFAR-100: the fraction of unsatisfied samples for various choices of the number of samples $m$. The embedding dimension is $d = 128$.

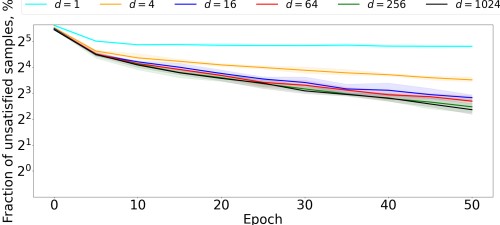
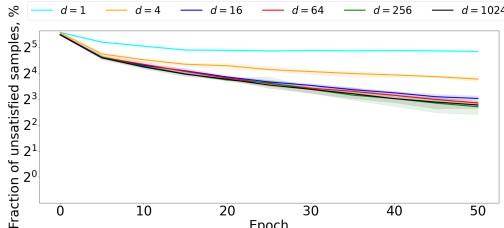

(c) CIFAR-10: the fraction of unsatisfied samples for various choices of the embedding dimension $d$. The number of samples $m = 10^5$.

(d) CIFAR-100: the fraction of unsatisfied samples for various choices of the embedding dimension $d$. The number of samples $m = 10^5$.

Figure 4: Experiments on CIFAR-10 (left) and CIFAR-100 (right). The data points show the average over 5 runs, and the shaded area shows the minimum and the maximum values over the runs

the resulting embedding is consistent with almost all ($\geq 99\%$) triplets. On the other hand, for $m \in \{10^5, 10^6\}$, $d$ is substantially less than $\sqrt{m}$, and the number of satisfied samples sharply drops from $99\%$ to $93\%$. This is consistent with our theoretical results in Theorem 1.

Not surprisingly, Figures 4c and 4d show that, when the embedding dimension increases, so does the accuracy, i.e. the number of satisfied triplets. But the accuracy stops increasing when the dimension reaches approximately $\sqrt{m} \approx 316$ – while there is a $2\%$ accuracy increase when the dimension changes from $64$ to $256$, there is no accuracy increase when the dimension changes from $256$ to $1024$. This again conforms with our result from Theorem 1.

## 5 Conclusion

In this paper, we provide bounds on the necessary and sufficient dimension to represent a collection of contrastive constraints of the form "distance from $x$ to $y$ is smaller than distance from $x$ to $z$". This is a fundamental question in machine learning theory, since it educates the choice of deep learning architectures by providing guidance for the size of the embedding layer. Our experiments illustrate the predictive power of our theoretical findings in the context of deep learning. We also believe that it gives rise to many interesting directions for future work depending on the exact desiderata: approximate versions, different choices of normed spaces, bi-criteria algorithms, agnostic settings.

While the considered distance comparison settings play a central role in contrastive learning and nearest neighbor search, so far there has been no theoretical studies of their embedding dimension. Our work is the first to present a series of such upper and lower bounds in a variety of settings via a novel connection to the notion of arboricity from graph theory. As a follow-up, one can consider an improved embedding construction for k-NN: in the upped bound from Section 3, the dependence on both $\log n$ and $k$ can likely can be improved. Another interesting direction is tighter data-dependent bounds on dimension: while we provide fine-grained bounds in terms of arboricity – which are potentially much stronger than bounds in terms of the number of edges – they don't necessary capture properties of dataset which can lead to sharper bounds.

## Acknowledgments and Disclosure of Funding

We would like to thank Michael Barash for several very helpful suggestions. Work by Orr Fischer was partially supported by the Israel Science Foundation (grant No. 1042/22 and 800/22). Work by Vaggos Chatziafratis was partially supported by Hellman's fellowship and startup grant at UC Santa Cruz.

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

# A   Additional experiments

|            | $m = 10^2$ | $m = 10^3$ | $m = 10^4$ | $m = 10^5$ | $m = 10^6$ | $m = 10^7$ |
|------------|------------|------------|------------|------------|------------|------------|
| $n = 128$  | 6, 8       | 32, 34     | 162, 169   | 256        | 256        | 256        |
| $n = 256$  | 6          | 19, 20     | 135, 139   | 452, 464   | 512        | 512        |
| $n = 512$  | 4, 6       | 12         | 80, 82     | 502, 508   | 1024       | 1024       |
| $n = 1024$ | 4, 6       | 8          | 43, 44     | 363, 366   | 1453, 1473 | 2048       |
| $n = 2048$ | 4, 6       | 6          | 24, 25     | 202, 204   | 1479, 1488 | 3931, 3971 |

Table 3: Embedding dimension based on construction from Section 2. For each pair of $n$ and $m$, we show the minimum and the maximum dimensions obtained over 10 runs (we show a single number when the minimum and the maximum are equal).

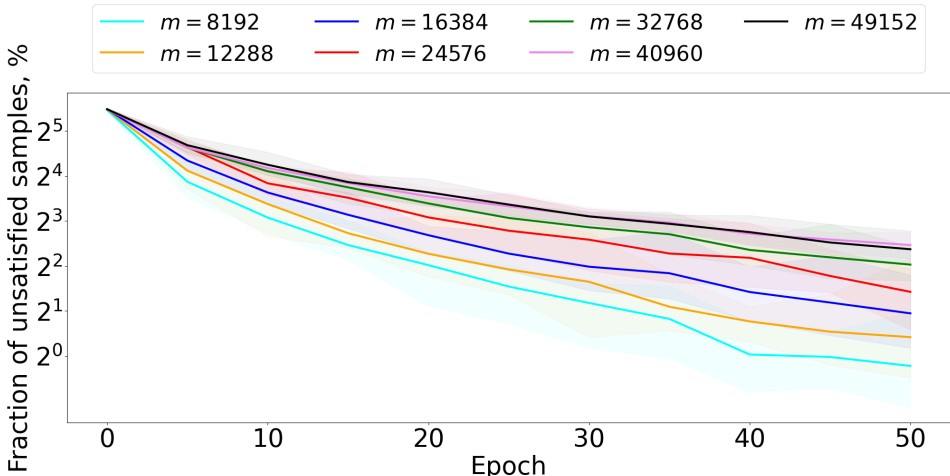

Figure 5: CIFAR-100: the fraction of unsatisfied samples for various choices of the number of samples $m$. The embedding dimension is $d = 128$.

In this section, we present additional experiments.

**Contrastive Samples**   In Table 3, for various values of $n$ and $m$, we show the dimensions of the embeddings constructed according to Section 2. We sample $m$ random triplets from the CIFAR-100 dataset, and label the triplets based on a ground-truth embedding generated using a pretrained ResNet18 network. Note that the embedding dimension is always at most $2n$, which corresponds to the case when the constraint graph is a clique. Moreover, in agreement with our theory, when $m < n^2$, increasing $n$ decreases the required dimension: the constraint graph becomes more sparse, which decreases the arboricity.

In Figure 5, similarly to Figure 4b, we show training accuracy on CIFAR-100 dataset for various values of $m$. In this figure, we focus on the setting when $m$ is close to $d^2 = 16384$. While for $m \leq d^2/2$ the accuracy is close to perfect (99%), the accuracy decreases starting from this point. This supports our theoretical result that $d = \Theta(\sqrt{m})$ dimensions are required to preserve the contrastive samples.

**$k$-NN**   In Table 4, we present results for $k$-NN settings for $d = 128$ and for various choices of $n$ and $k$. We sample $n$ points from the CIFAR-10 dataset, and generate $k$-NN based on a ground-truth embedding generated using a pretrained ResNet18 network. For each element $x$, let $\pi_1^*(x), ..., \pi_{n-1}^*(x)$ be the ordering of other elements according to the ground-truth embedding. For each and $i \in [k]$ and each $j > i$, we generate contrastive samples $(x, \pi_i^*(x)^+, \pi_j^*(x)^-)$, and we train the neural network on this set of samples similarly to Section 4.

For each $n$ and $k$, we report the loss function measuring the quality of preserving the $k$-NNs, defined as follows. For each vertex $x$ and each $i \in [k]$, we compute the change of rank of the $i$'th nearest neighbor of $x$ in the new embeddings. Formally, we find $j$ such that the $i$'th nearest neighbor of

| | $k=1$ | $k=2$ | $k=4$ | $k=8$ | $k=16$ | $k=32$ | $k=64$ |
|---|---|---|---|---|---|---|---|
| $n=10$ | 0.0, 1.8 | 0.05, 0.8 | 0.15, 0.8 | 0.22, 0.78 | | | |
| $n=100$ | 0.01, 0.07 | 0.21, 0.32 | 0.58, 0.7 | 1, 1.27 | 1.6, 1.8 | 2.16, 2.46 | 2.5, 2.9 |
| $n=1000$ | 0.04, 0.07 | 0.34, 0.4 | 0.81, 0.93 | 1.47, 1.54 | 2.3, 2.4 | 3.36, 3.44 | 4.7, 4.9 |

Table 4: Training loss for preserving $k$-NNs for various values of $n$ and $k$. For each pair of $n$ and $k$, we show the minimum and the maximum dimensions obtained over 10 runs (we show a single number when the minimum and the maximum are equal).

$x$ according to the ground-truth embedding is the $j$'th nearest neighbor according to the trained embedding, contributing $|i - j|$ to the loss. Finally, we define the final loss as the average loss over all $x \in V$ and $i \in [k]$.

Table 4 shows that the loss increases with both $k$ and $n$. However, dependence on $n$ is much lower than dependence on $k$, supporting our theoretical result which shows polynomial dependence on $k$ and only polylogarithmic dependence on $n$.

# B  Preliminaries

**Definition 22** (Metric, semimetric)**.** *An* metric space *is an ordered pair* $(X, \delta_X)$ *consisting of a set* $X$ *and a map* $\delta_X \colon X \times X \to [0, \infty]$ *such that* $\delta_X$ *satisfies:*

1. $\delta_X(x, y) = 0 \iff x = y$;

2. $\delta_X(x, y) = \delta_X(y, x)$, *for all* $x, y \in X$;

3. $\delta_X(x, y) + \delta_X(y, z) \leq \delta_X(x, z)$, *for all* $x, y, z \in X$.

*If* $\delta_X$ *satisfies the last two properties but only* $\delta_X(x, x) = 0$ *for all* $x \in X$ *instead of the first one then it is called a* semimetric.

We note that the triangle inequality doesn't affect our results. Intuitively, our goal is to preserve the ranking of distances, and adding a sufficiently large constant to distances preserves the ranking while also satisfying the triangle inequality.

**Definition 23** ($\ell_p$ norm, $\ell_p^p$ distance function, $\ell_\infty$ norm)**.** *Given vectors* $v, v' \in \mathbb{R}^d$ *and* $p \geq 1$, *the distance between* $v$ *and* $v'$ *under the* $\ell_p$ *norm is*

$$\delta_{\ell_p}(v, v') = \left( \sum_{i=1}^{d} \big| v[i] - v'[i] \big|^p \right)^{1/p},$$

*where* $v[i]$ *is the* $i$'th *coordinate of vector* $v$. *The distance between* $v$ *and* $v'$ *under* $\ell_p^p$ *is*

$$\delta_{\ell_p^p}(v, v') = \sum_{i=1}^{d} \big| v[i] - v'[i] \big|^p.$$

*The distance between* $v$ *and* $v'$ *under the* $\ell_\infty$ *norm is*

$$\delta_{\ell_\infty}(v, v') = \max_{i \in [d]} \big| v[i] - v'[i] \big|.$$

*For* $p \geq 1$ *or* $p = \infty$, *the norm of* $v$ *is defined as* $\|v\|_p = \delta_{\ell_p}(v, v)$.

**Fact 24** (Chernoff bound)**.** *Let* $X_1, \ldots, X_r \in \{0, 1\}$ *be mutually independent random variables. Denote by* $\mu = \mathbb{E}\left[\sum_{i=1}^{r} X_i\right]$, *the expectation of the sum of variables. Then for any* $0 < \gamma < 1$ *it holds that*

$$\mathbb{P}\left[ \left| \sum_{i=1}^{r} X_i - \mu \right| \geq \gamma \mu \right] \leq 2\exp\left( -\frac{\gamma^2 \mu}{3} \right).$$

*Additionally, for any* $\gamma > 0$ *it holds that*

$$\mathbb{P}\left[ \sum_{i=1}^{r} X_i \geq (1 + \gamma)\mu \right] \leq \exp\left( -\frac{\gamma^2 \mu}{2 + \gamma} \right).$$

## B.1 Ordinal Embeddings

**Fact 25** ([BL05]). *Given a set of non-contradictory inequalities among pairwise distances on $V$, there exists a metric $\delta\colon V \times V \to \mathbb{R}_{\geq 0}$ which satisfies all the inequalities.*

*Proof.* Consider a graph whose vertices are $V \times V$ and create a directed edge between two vertices if they participate in some inequality. Since the inequalities are non-contradictory, there are no cycles in this graph. Consider any topological ordering of this graph and define $w_{ij}$ to be the index of each pair in the topological ordering. Let $\delta = W + w_{ij}$ where $W = |V|^2$. Note that $\delta$ satisfies the triangle inequality. □

## B.2 Arboricity

In this subsection, we present basic facts about arboricity, and analyze the arboricity of the constraint graph in our various settings.

For a directed graph, we say that the out-degree of a vertex $x$ is $R$, for some integer $R$, if $x$ has $R$ incident edges oriented towards $x$.

**Fact 26** ([AMR92], Lemma 2.2). *If the edges of $G$ can be oriented such that each vertex has in-degree at most $R$ for some integer $R$, then $r \leq R + 1$.*

**Lemma 27.** *The constraint graph in the* k-NN *setting has arboricity at most $k + 1$.*

*Proof.* In the constraint graph of a k-NN instance, we have an edge for each pair $(x, \pi_i(x))$ for $1 \leq i \leq k, x \in V$, where $\pi_i$ is the $i$'th nearest neighbor of $x$. If for each such pair, we orient the edge inwards to $x$, we obtain a directed graph with in-degree at most $k$. Therefore, by Fact 26, the constraint graph $G$ has arboricity at most $k + 1$. □

Finally, the following fact relates the number of edges $m$ and the arboricity of the graph.

**Lemma 28** ([DHS91] Theorem 2). *Any graph $G$ with $m$ edges has arboricity $r \leq \lceil \sqrt{m/2} \rceil$.*

# C Missing Proofs From Sections 2 and 3

*Proof of Lemma 11.* Since $|N^-(x)| \leq h$ for all $x$, it suffices to show that with probability 1, any subset $A \subseteq \{\hat{x}_1, \ldots, \hat{x}_n\}$ of size $|A| \leq h$ is linearly independent. We prove it by induction on $|A|$, and the base case $|A| = 0$ trivially holds.

For the induction step, let $\hat{x}$ be the last point in $A$. By the induction hypothesis, $A \setminus \{\hat{x}\}$ are linearly independent. Let $H = \text{Span}(A \setminus \{\hat{x}\})$, and let $B$ the ball where $\hat{x}$ is sampled from. Since $\text{Vol}(H) = 0$, and $\text{Vol}(B) > 0$, we have $\mathbb{P}[\hat{x} \in H] = 0$, meaning that $A$ are linearly independent. □

## C.1 Proof of Lemma 21

In this section, $\delta(x, y) = \|x - y\|_1$. For other $\ell_p$ for $p \in [1, +\infty)$, the construction is the same by replacing each coordinate value $c$ with $c^{1/p}$.

**Agreement sets**   Before proving Lemma 21, we define the following sets:

$$\text{Agr}(x, y) = \{j \in [L] \mid \hat{x}^{(j)} = \hat{y}^{(j)}\},$$
$$\text{Agr}_{\text{D}}(x, y) = \{j \mid \hat{x}^{(j)} = \hat{y}^{(j)} = I_x \text{ or } \hat{x}^{(j)} = \hat{y}^{(j)} = I_y\},$$
$$\text{Agr}_{\text{N}}(x, y) = \{j \mid \exists z \in N^-(x) \cap N^-(y) \text{ such that } \hat{x}^{(j)} = \hat{y}^{(j)} = I_z\}.$$

The idea is to measure on which indices the points agree and to differentiate sources of agreements.

- $\text{Agr}(x, y)$ is the set of indices on which $x$ and $y$ agree, i.e. choose the same vector.
- $\text{Agr}_{\text{D}}(x, y)$ is the set of indices where $x$ and $y$ choose the same vector by a direct connection: $x$ chooses its own set $I_x$ and $y$ chooses $x$'s set $I_x$ (or reverse).

- $\mathrm{Agr_N}(x, y)$ is the set of indices where $x$ and $y$ choose the same vector by an indirect connection: $x$ and $y$ share a common neighbor $z$, and both choose $z$'s set $I_z$.

Note that $\mathrm{Agr} = \mathrm{Agr_D} \cup \mathrm{Agr_N}$. When $x$ and $y$ are neighbors and $x'$ and $y$ are not, we will show that $|\mathrm{Agr}(x, y)| \approx |\mathrm{Agr_D}(x, y)| \gg |\mathrm{Agr}(x', y')|$.

**Choice of parameters**   We next remind the choice of parameters.

| | |
|---:|:---|
| Graph arboricity | $r \leq 2k$ |
| The size of the design | $R = \Theta(r^3 \log^3 n)$ |
| Probability that an element doesn't choose its own design vector | $p = O(1/(r \log n))$ |
| Probability that neighbors choose the same design vector | $\gamma = \frac{p(1-p)}{2r} = \Theta(\frac{1}{r^2 \log n})$ |
| Fraction of intersecting elements between sets in the design | $\alpha = \Theta(\frac{1}{r^3 \log^2 n})$ |
| The number of blocks corresponding to designs | $L = \Theta(r^4 \log^4 n)$ |

We next justify the choice of the parameters.

- In the proof of Theorem 5, to counter the $K = O(r \log n)$ term from Section 3.1, for $\{x, y\} \in E$ we need to bound the spread of $|\mathrm{Agr}|$ *for neighbors* as $\frac{|\mathrm{Agr}|}{r \log n}$. To achieve that, we need to bound the spread of both $|\mathrm{Agr_N}|$ and $|\mathrm{Agr_D}|$.

- First, we need to guarantee that $|\mathrm{Agr_N}| = O\left(\frac{|\mathrm{Agr_D}|}{r \log n}\right)$. In Proposition 30, we require that $2r\left(\frac{p}{2r}\right)^2 \leq \frac{c\gamma}{4r \log n}$. This is since $2r\left(\frac{p}{2r}\right)^2$ bounds the probability that two points select the same common neighbor (which counts towards $\mathrm{Agr_N}$), while $\gamma$ is the probability that $x$ will choose $I_y$ for $y \in N^-(x)$ (which counts towards $\mathrm{Agr_D}$). Since $\gamma = \frac{p(1-p)}{2r}$, this bounds $p = O(1/(r \log n))$ and $\gamma = O(1/(r^2 \log n))$.

- To bound the spread of $|\mathrm{Agr_D}|$, note that $\mathbb{E}[|\mathrm{Agr_D}|] = \gamma L$. To bound the spread as $\frac{|\mathrm{Agr_D}|}{r \log n}$, by Chernoff we must have $\gamma L = r^2 \log^3 n$, meaning $L = r^4 \log^4 n$.

- Different sets from the design (Definition 19) intersect by at most $\alpha R$ elements. When two points sample different sets, the distance between their embeddings increases by the number of elements outside of their intersection, which is at least $2R - \alpha R$. Note that the distance between neighbors will be approximately

$$(2R - \alpha R)(L - |\mathrm{Agr_D}|) \approx 2RL(1 - \frac{\alpha}{2} - \gamma(2 - \alpha))$$

Similarly to the above, we want to bound the deviation due to the $\alpha/2$ term, and by the same logic we choose $\alpha = O(\gamma/(r \log n)) = O(1/(r^3 \log^2 n))$.

- We need to choose $2n$ sets from the design. Since by Lemma 20 the design has $2^{\alpha R \log R}$ sets, to guarantee that this value is at least $2n$, we take $\alpha$ and $R$ so that $\alpha R = \Omega(\log n)$, meaning $R = \Theta(r^3 \log^3 n)$.

**Proofs**   The next statement shows concentration of $\mathrm{Agr_D}$ for neighbors and non-neighbors.

**Proposition 29.** *For any $x, y \in V$, if $\{x, y\} \notin E$ then $|\mathrm{Agr_D}(x, y)| = 0$, and if $\{x, y\} \in E$, then $\left||\mathrm{Agr_D}(x, y)| - \gamma L\right| \leq \frac{c\gamma L}{4r \log n}$ w.h.p.*

*Proof of Proposition 29.* If $\{x, y\} \notin E$, then $x \notin N^-(y)$ and $y \notin N^-(x)$, i.e. for every $j \in [L]$, we have $\hat{x}^{(j)} \neq I_y$ and $\hat{y}^{(j)} \neq I_x$ and hence $|\mathrm{Agr_D}(x, y)| = 0$.

If $\{x, y\} \in E$, assume w.l.o.g. that $y \in N^-(x)$. Therefore, $j \in \mathrm{Agr_D}(x, y)$ if and only if we set $\hat{y}^{(j)} = I_y$ and $\hat{x}^{(j)} = I_y$. Recall that $\mathbb{P}\left[\hat{y}^{(j)} = I_y\right] = 1 - p$ and $\mathbb{P}\left[\hat{x}^{(j)} = I_y\right] = p/(2r)$.

Therefore,

$$\mathbb{P}[j \in \mathrm{Agr_D}(x, y)] = \frac{p(1-p)}{2r} = \gamma,$$

i.e. $\mathbb{E}[|\mathrm{Agr_D}(x, y)|] = \gamma L$. Since $\gamma L = \Omega(r^2 \log^3 n)$, by Chernoff, w.h.p. we have

$$\left||\mathrm{Agr_D}(x, y)| - \gamma L\right| \leq \frac{c\gamma L}{4r \log n} \qquad \square$$

We next show concentration for $\mathrm{Agr_N}$. Note that $\mathrm{Agr_D}$ for neighbors is much larger than $\mathrm{Agr_N}$ for both neighbors and non-neighbors.

**Proposition 30.** *For any $x, y \in V$, we have $0 \leq |\mathrm{Agr_N}(x,y)| \leq \frac{c\gamma L}{4r \log n}$ w.h.p.*

*Proof.* We bound the expectation of $|\mathrm{Agr_N}(x,y)|$. Recall that $j \in \mathrm{Agr_N}(x,y)$ if and only if there exists a point $z \in N^-(x) \cap N^-(y)$ such that $\hat{x}^{(j)} = I_z$ and $\hat{y}^{(j)} = I_z$. Moreover, the events $\hat{x}^{(j)} = I_z$ and $\hat{y}^{(j)} = I_z$ are independent, each occurring with probability $p/2r$.

Since $|N^-(x) \cap N^-(y)| \leq |N^-(x)| \leq 2r$, we have

$$\mathbb{E}\left[|\mathrm{Agr_N}(x,y)|\right] \leq |N^-(x)| \cdot L \left(\frac{p}{2r}\right)^2 = \frac{Lp^2}{2r}.$$

Finally, we note that $\frac{Lp^2}{2r} = \Omega(\log n)$, and by Chernoff, w.h.p.:

$$|\mathrm{Agr_N}(x,y)| \leq \frac{4}{3}\mathbb{E}\left[|\mathrm{Agr_N}(x,y)|\right] = \frac{4}{3} \cdot \frac{Lp^2}{2r} \leq \frac{c\gamma L}{4r \log n} \qquad \square$$

*Proof of Lemma 21.* Recall that $\delta(\hat{x}, \hat{y}) = \sum_{j=1}^{L} \delta(\hat{x}^{(j)}, \hat{y}^{(j)})$. For each $j \in \mathrm{Agr}(x,y)$, we have $\delta(\hat{x}^{(j)}, \hat{y}^{(j)}) = 0$, and for $j \notin \mathrm{Agr}(x,y)$, due to the property of the $(\alpha, R)$-design, we have

$$|\delta(\hat{x}^{(j)}, \hat{y}^{(j)}) - 2R| \leq 2\alpha R.$$

Summing over all $j \notin \mathrm{Agr}(x,y)$, we get

$$\left|\delta(\hat{x}, \hat{y}) - 2(L - |\mathrm{Agr}(x,y)|)R\right| \leq 2\alpha RL \leq \frac{c\gamma}{4r \log n} RL, \qquad (2)$$

where we used $\alpha \leq \frac{c\gamma}{8r \log n}$.

**Non-neighbors** If $\{x,y\} \notin E$, then by Propositions 29 and 30 we have $0 \leq |\mathrm{Agr}(x,y)| \leq \frac{c\gamma L}{4r \log n}$. By Equation (2) we have

$$\left|\delta(\hat{x}, \hat{y}) - 2\left(1 - \frac{c\gamma}{4r \log n}\right) RL\right| \leq \frac{c\gamma}{4r \log n} RL \implies |\delta(\hat{x}, \hat{y}) - 2RL| \leq \frac{c\gamma RL}{r \log n}$$

**Neighbors** If $\{x,y\} \in E$, then by Propositions 29 and 30,

$$||\mathrm{Agr}(x,y)| - \gamma L| \leq \frac{2c\gamma L}{4r \log n}$$

and by Equation (2) it follows that

$$|\delta(\hat{x}, \hat{y}) - 2\left(1 - \gamma\right) RL| \leq 2\left(\frac{2c\gamma}{4r \log n}\right) RL \leq \frac{c\gamma RL}{r \log n}. \qquad \square$$

# D   Contrastive Queries in $\ell_p$ Norm

In this section, we show upper bounds for dimensions for embedding into space with $\ell_p$-norms or $\ell_\infty$-norm.

## D.1   Contrastive Queries for Finite $p$

In this section, we prove Theorem 3, which provides an upper bound of $m + 1$ on the embedding dimension in $\ell_p$ for integer $p \geq 1$.

We say that a set $S \subseteq V \times V$ is *symmetric* if $(x, y) \in S \Leftrightarrow (y, x) \in S$.

Due to the symmetry, with a slight abuse of notation, we define the cardinality $|S|$ for symmetric sets to be equal to the number of distinct unordered pairs in $S$.

**Definition 31** (Partial semimetric). *An ordered triple $(V, S, \delta_S)$ consisting of a set $V$, a symmetric set $S \subseteq V \times V$ and a map $\delta_S \colon S \to [0, \infty)$ is a* partial semimetric space *if $\delta_S$ satisfies the following:*

1. *For all $x \in V$, if $(x, x) \in S$ then $\delta_S(x, x) = 0$.*

2. *$\delta_S(x, y) = \delta_S(y, x)$ for all $(x, y) \in S$,*

3. *$\delta_S(x, y) + \delta_S(y, z) \leq \delta_S(x, z)$ for all $(x, y), (x, z), (y, z) \in S$.*

**Definition 32** (Partial embedding). *We say that a partial semimetric $(V, S, \delta_S)$ partially embeds into a metric space $(Y, \delta_Y)$ if there exists a map $F \colon V \to Y$ such that $\delta_S(x, y) = \delta_Y(F(x), F(y))$ for all $(x, y) \in S$.*

The following lemma is an extension of the standard embedding result into $\ell_p$ (see e.g. [DL97]).

**Lemma 33.** *Let $\mathbf{S} = (V, S, \delta_S)$ be a partial semimetric on $V$ and let $m = |S|$. If $\mathbf{S}$ partially embeds into an $\ell_p^p$-space with finite dimension, then it embeds into $(\ell_p^p)^{m+1}$.*

*Proof.* Let $\{\{x_i, y_i\}\}_{i=1}^m$ be the unordered pairs of $S$. We assign every partial semimetric $(V, S, \delta)$ on $S$ an $m$-dimensional vector $v_\delta$, where $v_\delta[i] = \delta(x_i, y_i)$. We call $v_\delta$ the *representation vector* of $(V, S, \delta)$. Define $\mathrm{NOR}^S$ to be the set of representations of all partial semimetrics on $S$ which can be partially embedded into $\ell_p^p$, i.e.

$$\mathrm{NOR}^S = \{v_\delta \mid \text{There exists } d \in \mathbb{N} \text{ such that } (V, S, \delta) \text{ partially embeds into } (\mathbb{R}^d, \ell_p^p)\}.$$

Note that $\mathrm{NOR}^S$ is a cone:

1. If $v_\delta \in \mathrm{NOR}^S$ then $\alpha v_\delta \in \mathrm{NOR}^S$ for all $\alpha \geq 0$.

2. If $v_\delta, v_{\delta'} \in \mathrm{NOR}^S$ then $v_\delta + v_{\delta'} \in \mathrm{NOR}^S$.

An *extreme ray* is a point $v_\delta \in \mathrm{NOR}^S$ such that if $v_\delta = v_{\delta_1} + v_{\delta_2}$ for $v_{\delta_1}, v_{\delta_2} \in \mathrm{NOR}^S$ then it has to be that $v_{\delta_1} = \alpha v_\delta$ and $v_{\delta_2} = (1 - \alpha) v_\delta$ for some $\alpha \in [0, 1]$.

Next, we show that any extreme ray of $\mathrm{NOR}^S$ has a partial embedding into the one-dimensional space $(\mathbb{R}, \ell_p^p)$. Indeed, let $v_\delta$ be an extreme ray, and let $d$ be the minimum dimension for which $(V, S, \delta)$ partially embeds to $(\mathbb{R}^d, \ell_p^p)$. If $d = 1$, then we are done; otherwise, assume by contradiction that $d > 1$. Let $F : V \to \mathbb{R}^d$ such that $\delta(x, y) = \delta_p^p(F(x), F(y))$ for all $(x, y) \in S$. Let $F_1 : V \to \mathbb{R}, F_2 : V \to \mathbb{R}^{d-1}$ such that

$$F_1(x) = F(x)[1] \text{ and } F_2(x) = (F(x)[2], \ldots, F(x)[d]),$$

i.e. $F_1$ is the embedding $F$ restricted to the first dimension, and $F_2$ is $F$ restricted to the remaining $d - 1$ dimensions. We notice that for each $(x, y) \in V \times V$, $\delta_p^p(F(x, y)) = \delta_p^p(F_1(x, y)) + \delta_p^p(F_2(x, y))$.

Define $\rho_1, \rho_2 : S \to \mathbb{R}$ such that $\rho_1(x, y) = \delta_p^p(F_1(x), F_1(y))$, and $\rho_2(x, y) = \delta_p^p(F_2(x), F_2(y))$. Therefore, $v_\delta = v_{\rho_1} + v_{\rho_2}$. Since $v_\delta$ is an extreme ray, then there exists $\alpha \in [0, 1]$ such that $v_\delta = \alpha v_{\rho_1}$. In particular, $\delta$ can be partially embedded into one dimension, by taking the embedding $\alpha F_1(x)$, contradicting minimality of $d$. We conclude that $d = 1$.

Finally, let $v_\mathbf{S}$ be the representation vector of $\mathbf{S}$. By Caratheodory's theorem, since $v_\mathbf{S} \in \mathrm{NOR}^S$, there exists $m + 1$ extreme rays $v_{\delta_1}, \ldots, v_{\delta_{m+1}} \in \mathrm{NOR}^S$ such that $v_\mathbf{S} = \sum_{i=1}^{m+1} v_{\delta_i}$. We have shown that for each $i \in [m + 1]$, the partial semi-metric $(X, S, \delta_i)$ has a partial embedding $F^{(i)} : V \to \mathbb{R}$ into the one dimensional space $(\mathbb{R}, \ell_p^p)$. It follows that the embedding $F = (F^{(1)}, \ldots, F^{(m+1)})$ is a partial embedding of $\mathbf{S}$ into $(\mathbb{R}^{m+1}, \ell_p^p)$, and the claim follows. $\square$

*Proof of Theorem 3.* If $Q$ is a set of non-contradictory constraints, then we can embed it into $\ell_2$ using Theorem 1. We can then embed it isometrically into $\ell_p$ (see Chapter 1.5 from [Mat13] and Theorem 5 from [Goe06]). By using the same points, the relationships between distances are also preserved in $\ell_p^p$. Let $S$ be the set of all edges in the constraint graph $G$. Then we have a partial semimetric $(V, S, \delta_S)$ which is partially embedded into $\ell_p^p$. By Lemma 33 it partially embeds isometrically into $(\ell_p^p)^{|S|+1}$. For the same embedding, the relationships between distances are also preserved in $\ell_p$. $\square$

### D.2 Contrastive Queries in $\ell_\infty$ Norm

In this section, we prove Theorem 2, which states that dimension $O(m^{2/3})$ suffices to satisfy any set of $m$ non-contradictory contrastive queries $Q$ in the $\ell_\infty$ norm.

Let $G = (V, E)$ be the constraint graph, where $E$ is the edge set. We arbitrarily assign a unique identifier $\mathrm{id}(x) \in [n]$ for each $x \in V$. Let $V_{\mathrm{high}} \subseteq V$ be the set of points with degree with at least $m^{1/3}$ in $G$. Let $V_{\mathrm{low}} = V \setminus V_{\mathrm{high}}$.

Our embedding is a concatenation of two embeddings $F_1$ and $F_2$, which intuitively "handle" $V_{\mathrm{low}}$ and $V_{\mathrm{high}}$ respectively. In the sub-embedding $F_1$, we use the fact that the graph induced by $V_{\mathrm{low}}$ has low degree to argue that it has a proper *distance-2-edge coloring* with $O(m^{2/3})$ colors, i.e. we can color the edges of the graph such that no two edges at distance at most 2 share the same color. We use this coloring to obtain an embedding $F_1 \colon V_{\mathrm{low}} \to \mathbb{R}^{O(m^{2/3})}$ which satisfies certain distance properties between any pair of neighbors in $V_{\mathrm{low}}$. We then extend $F_1$ to an embedding $F \colon V \to \mathbb{R}^{O(m^{2/3})}$ which is consistent with $Q$. This extension draws inspiration from the seminal Fréchet embedding [Fré10]: for each point in $x_i \in V_{\mathrm{high}}$ we add a single distinct dimension $i$, in which we intuitively set this coordinate for each point $x \in V$ as distance from $x_i$ in $F'$. In actuality, we set these coordinates slightly differently, in order to combine correctly with the sub-embedding $F_1$, and obtain an embedding which is consistent with $Q$. By Lemma 34, the size $|V_{\mathrm{high}}| = O(m^{2/3})$, which implies that together the dimension of $F$ is $O(m^{2/3})$.

**Lemma 34.** *Let $Q$ be the set of $m$ contrastive queries. Let $V_{\mathrm{high}}$ be the set of points with degree at least $m^{1/3}$ in the constraint graph. Then $|V_{\mathrm{high}}| = O(m^{2/3})$.*

*Proof.* Recall that each query $(x, y^+, z^-) \in Q$ is associated with two edges, $\{x, y\}, \{x, z\} \in E$. Hence, the total number of edges in $G$ is at most $2|Q| = 2m$. This implies

$$m^{1/3}|V_{\mathrm{high}}| \leq \sum_{x \in V_{\mathrm{high}}} \deg(x) \leq \sum_{x \in V} \deg(x) = 2|E| = 4m.$$

By rearrangement, we obtain that $|V_{\mathrm{high}}| \leq 4m^{2/3}$. $\qquad\square$

Since $Q$ is non-contradictory, by Fact 25 there exists a metric $\delta$ consistent with $Q$. Using the Frechét embedding [Fré10], any metric on $n$ points may be isometrically embedded into $\mathbb{R}^{n-1}$ under the $\ell_\infty$ norm.

**Definition 35** (Scaled Fréchét embedding $F'$). *Let $F' \colon V \to \mathbb{R}^{n-1}$ be an embedding of $\delta$ into the cube $[0, 1/2]^{n-1}$ under the $\ell_\infty$ norm, obtained by scaling and shifting (i.e. multiplying or adding some value to all coordinates, respectively) the Fréchét embedding of $\delta$.*

We note that scaling and shifting do not affect whether a contrastive query is satisfied, therefore $F'$ is consistent with $Q$ as well.

**Lemma 36.** *There is an embedding $F_1$ of $V_{\mathrm{low}}$ into $\mathbb{R}^{O(m^{2/3})}$ such that the following hold:*

*(a) for each $x \in V_{\mathrm{low}}$ and $i \in \mathbb{N}$, it holds that $F_1(x)[i] \in [0, 1]$;*

*(b) for each $x, y \in V_{\mathrm{low}}$ such that $\{x, y\} \in E$, it holds that $\|F_1(x) - F_1(y)\|_\infty = 1/2 + \|F'(x) - F'(y)\|_\infty$.*

*Proof.* By definition, each $x \in V_{\mathrm{low}}$ has degree at most $\Delta = O(m^{1/3})$. Therefore, there is an edge coloring $C \colon E \to [\Delta^2 + 2]$ of $G = (V, E)$, in which (a) every vertex has at most one incident edge of any color, and (b) any two adjacent vertices $x, y$ share exactly one edge color – the one of their shared edge $C(x, y)$. We remark that this coloring is called in the literature *distance-2-edge coloring*. Such a coloring can be found using a greedy approach, where we color the edges one by one, where for each edge $\{x, y\}$ we choose a color that is not taken by previous edges of $x, y$ or by edges of any neighbor $z \in N(x) \cup N(y)$. In other words, let $K(x, y)$ be the set of colors taken by any edge incident to any vertex in $\{x, y\} \cup N(x) \cup N(y)$. Since $|K(x, y)| \leq 2\Delta^2 + 1$, then we can always choose from $\{x, y\}$ a color different from all colors of $K(x, y)$.

We define the embedding $F_1$ as follows: assume the color of the edge of $\{x, y\}$ is $C(x, y) \in [\Delta^2 + 2]$. Let $c = C(x, y)$ and assume w.l.o.g. that $\mathrm{id}(x) < \mathrm{id}(y)$. We define $F_1(x)[c] = 0$ and $F_1(y)[c] = 1/2 + \|F'(x) - F'(y)\|_\infty$. For any $x \in V$, if a coordinate $i$ is not set in this process, we set $F_1(x)[i] = 1/2$. We note this is well-defined since the edge coloring is proper (i.e. no vertex has two edges of the same color), so no coordinate is set twice. This concludes the description of the embedding.

Next, we show that properties (a) and (b) hold. Recall that we consider distances over $\ell_\infty$, hence for each pair $x, y \in V_{\mathrm{low}}$ there is a coordinate $i(x, y) \in \mathbb{N}$ for which $\|F'(x) - F'(y)\|_\infty = \|F'(x)[i(x, y)] - F'(y)[i(x, y)]\||$.

For property (a), we note that every coordinate $i$ is either set to $F(y)[i] = 0$, or to $F(y)[i] = 1/2 + \|F'(x) - F'(y)\|_\infty$ for some $x \in V$. Since $\|F'(x) - F'(y)\|_\infty \in [0, 1/2]$, and hence $\|F(x) - F(y)\|_\infty = 1/2 + \|F'(x) - F'(y)\|_\infty \in [0, 1]$, property (a) follows.

Next, we show that property (b) holds. Denoting $c = C(x, y)$, for each edge $\{x, y\} \in E$ such that $\mathrm{id}(x) < \mathrm{id}(y)$, it holds that $F'(x)[c] = 0$ and $F'(y)[c] = 1/2 + \|F(x) - F(y)\|_\infty$. Second, since $x, y$ share only one edge color, in each other coordinate $j \neq c$, either $F_1(x)[j] = 1/2$ or $F_1(y)[j] = 1/2$, meaning that $c$ is the coordinate with the maximum difference, i.e. $\|F_1(x) - F_1(y)\|_\infty = 1/2 + \|F'(x) - F'(y)\|_\infty$. $\qquad\square$

**The Overall Embedding:**

**Lemma 37.** *Let $F_1 \colon V_{\mathrm{low}} \to \mathbb{R}^{O(m^{2/3})}$ be the embedding described in Lemma 36. Then there exists an embedding $F \colon V \to \mathbb{R}^{O(m^{2/3})}$ such that for any $\{x, z\} \in E$ it holds that $\|F(x) - F(z)\|_\infty = 1/2 + \|F'(x) - F'(z)\|_\infty$.*

*Proof.* Let $d = O(m^{2/3})$ be the dimension of $F_1$ (i.e. $F_1 \colon V_{\mathrm{low}} \to \mathbb{R}^d$), and $r = |V_{\mathrm{high}}| = O(m^{2/3})$. Let $V_{\mathrm{high}} = \{y_1, \ldots, y_r\}$. We define $F(x)$ as follows: for $y_i \in V_{\mathrm{high}}$, we set all coordinates for $1 \leq j \leq (d + i - 1)$ to $F(y_i)[j] = 1/2$, the $(d+i)$'th coordinate as $F(y_i)[d+i] = 0$, and set any coordinate $d + i + 1 \leq j \leq d + r$ to $F(y_i)[j] = 1/2 + \|F'(y_i) - F'(y_{j-d})\|_\infty$. For $x \in V_{\mathrm{low}}$, we set the first $d$ coordinates to be as in $F_1(x)$. The remaining $r$ coordinates, i.e. $d + 1 \leq j \leq d + r$, we define as $F(x)[j] = 1/2 + \|F'(x) - F'(y_j)\|_\infty$. This concludes the description of the embedding.

First, we show that for any $x, z \in V_{\mathrm{low}}$ such that $\{x, z\} \in E$, it holds that $\|F(x) - F(z)\|_\infty = 1/2 + \|F'(x) - F'(z)\|_\infty$. This indeed holds by Lemma 36, and by the fact that in all the $r$ last coordinates are set to a value in $[1/2, 1]$, i.e. the difference on any of these coordinates is at most $1/2$.

Next, we show that for any $x \in V_{\mathrm{low}}$ and $y_i \in V_{\mathrm{high}}$ such that $\{x, y_i\} \in E$, it holds that $\|F(x) - F(y_i)\|_\infty = 1/2 + \|F'(x) - F'(y_i)\|_\infty$. Indeed, in any coordinate $j \neq (d + i)$, $F(y_i)[j] \geq 1/2$, and hence $|F(y_i)[j] - F(x)[j]| \leq 1/2$. On the other hand, in the $(d + i)$'th coordinate $|F(y_i)[d + i] - F(x)[d + i]| = 1/2 + \|F'(x) - F'(y_i)\|_\infty > 1/2$.

Finally, we consider the case where $y_i, y_j \in V_{\mathrm{high}}$ such that $\{y_i, y_j\} \in E$ and $i < j$. For the first $d$ coordinates, both vectors are set to $1/2$, in all coordinates between $d + 1, \ldots, d + j - 1$ the vector $y_j$ is set to $1/2$, and therefore they differ by at most $1/2$ in these coordinates. For the $(d + j)$'th coordinate, $y_j$ is set to zero, and $y_i$ is set to $1/2 + \|F'(y_i) - F'(y_j)\|_\infty$. For higher coordinates, both $y_i, y_j$ are set to values at least $1/2$. Therefore, $\|F(y_i) - F(y_j)\|_\infty = 1/2 + \|F'(y_i) - F'(y_j)\|_\infty$. $\qquad\square$

Finally, we show that $F$ is consistent with $Q$.

**Lemma 38.** *The embedding $F \colon V \to \mathbb{R}^{O(m^{2/3})}$ is consistent with $Q$.*

*Proof.* For any $(x, y^+, z^-) \in Q$, it holds that $\|F'(x) - F'(y)\|_\infty < \|F'(x) - F'(z)\|_\infty$, and therefore

$$\|F(x) - F(z)\|_\infty = 1/2 + \|F'(x) - F'(z)\|_\infty > 1/2 + \|F'(x) - F'(y)\|_\infty = \|F(x) - F(y)\|_\infty.$$

And since $\|F(x) - F(y)\|_\infty < \|F(x) - F(z)\|_\infty$, the query $(x, y^+, z^-)$ is satisfied. $\qquad\square$

Theorem 2 follows directly from Lemma 38.

# E   Lower Bounds

In this section, we prove lower bounds for all our settings. Before presenting the main theorem of this section, we formally introduce the notion of ordinally embedding a metric $\delta$ into $\ell_p$ space.

Recall that for $x \in V$, we denote $\pi_1(x), \ldots, \pi_{n-1}(x)$ to be the points in $V \setminus \{x\}$ ordered by their distance from $x$.

**Definition 39** (Ordinal Embedding). *Given a metric $\delta$, the full ordinal sample set $Q(\delta)$ is the following set of samples: $Q(\delta) = \{(x, \pi_i^+(x), \pi_{i+1}^-(x)) \mid x \in V, i \in [n-2]\}$. We say that $\delta$ can be ordinally embedded in $\ell_p$ space in dimension $d$ if its full ordinal sample set $Q$ is consistent with some embedding in $\ell_p$ space with dimension $d$.*

Next, we present the main theorem of this section, from which we can obtain lower bounds for all our settings:

**Theorem 40.** *For $p \in \mathbb{N} \cup \{\infty\}$, there exists a metric $\delta$ on $n$ points which can only be ordinally embedded in $\ell_p$-space using $d = \Omega(n)$ dimensions if $p$ is a constant even integer $p \geq 2$, or $d = \Omega(n/\log n)$ if $p$ is a constant odd integer $p \geq 1$ or $p = \infty$.*

We remark that the special case of $p = 2$ was previously proven in [CI24]. To prove Theorem 40, we need several propositions.

For a set of unlabeled triplets $C$, we say that a set of samples $Q$ is a labeling of $C$ if $Q$ has exactly one labeling for each unlabeled triplet of $C$ (and no other sample). We next show that there exists a set of $\Theta(n^2)$ triplets so that any its labeling is valid.

**Lemma 41** ([AAE$^+$24]). *For $V = \{x_1, \ldots, x_n\}$, let $C = \{(x_i, x_j, x_{j+1})\}_{1 \leq i < j < n}$ be the set of unlabeled triplets, whose labeling compares distances between $(x_i, x_j)$ and $(x_i, x_{j+1})$. Then for any labeling $Q$ of $C$, there is a metric $\delta_Q$ consistent with $Q$.*

*Proof.* Let $Q$ be a labeling of $C$. Fix anchor $x_i$ and consider a graph where we create a directed edge $x_j \to x_{j+1}$ when $(x_i, x_j^+, x_{j+1}^-) \in Q$, and an edge $x_{j+1} \to x_j$ when $(x_i, x_{j+1}^+, x_j^-) \in Q$. Note that for any $Q$ this graph is acyclic (since the corresponding undirected edges form a path), and hence there exists a topological sort $p_i$ on $x_{i+1}, \ldots, x_n$. We define a metric $\delta$ so that $\delta(x_i, x_j) = \delta(x_j, x_i) = n + p_i(x_j)$ for $i < j$ and $\delta(x_i, x_i) = 0$ for all $i$.

Note that $\delta$ is a metric: by construction, $\delta$ is symmetric and $\delta(x, x) = 0$ for all $x$, and the triangle inequality is satisfied since all distances are between $n$ and $2n$. Finally, note that $\delta$ satisfies all samples from $Q$. $\square$

Next, we use a claim proven in [AAE$^+$24], showing that any sufficiently large set of unlabeled triplets has an labeling which does not have a $d$-dimensional $\ell_p$ space embedding consistent with it (where the size of the unlabeled set is at least some function of $n, d, p$).

**Fact 42** ([AAE$^+$24], Reformulated). *Let $d$ be an integer, $V$ be a set of $n$ points, and $p \in \{1, 2, \ldots\} \cup \{\infty\}$ be constant. Then there exists a constant $c_p > 0$ such that for any sufficiently large $n$ the following hold.*

- *If $p$ is odd or $p = \infty$, then for any set of triplets $C$ of size at least $c_p nd \log n$ on $V$, there exists a labeling of $C$ which is not consistent with any $d$-dimensional $\ell_p$ space.*

- *If $p$ is even, then for any set of triplets $C$ of size at least $c_p nd$ on $V$, there exists a labeling which is not consistent with any $d$-dimensional $\ell_p$ space.*

*Proof of Theorem 40.* We consider the case of even $p$ – cases of odd and infinite $p$ are analogous. By Lemma 41, for some constant $c > 0$ there exists a set of triplets $C$ of cardinality at least $cn^2$ so that any labeling of $C$ is realizable by some metric. On the other hand, by Fact 42, when $|C| > c_p nd$, there exists a labeling $Q$ of $C$ which is not consistent with any $d$-dimensional $\ell_p$ space metric. Solving for $d$, unless $d > nc/c_p$, there exists a labeling which is not realizable in the $d$-dimensional $\ell_p$ space. Hence, $d = \Omega(n)$ for even $p$. $\square$

Next, we show lower bounds for our settings, namely for contrastive learning and k-NN, and for the extended settings of $t$-negatives and $t$-orderings. All lower bounds follow as immediate corollaries of Theorem 40.

**Theorem 43.** *Let $p$ be a positive even integer.*

1. *(Contrastive triplets) There exists a set of non-contradictory triplet samples $Q$ of size $|Q| = m$ for which any embedding in $\ell_p$ space consistent with $Q$ must have $d = \Omega(\sqrt{m})$.*

2. *($t$-negatives) There exists a set of non-contradictory $t$-negatives samples $Q$ of size $|Q| = m$ such that any embedding in $\ell_p$ space in $d$ dimensions requires $d = \Omega(\sqrt{m})$.*

3. *($t$-orderings) There exists a set of non-contradictory $t$-ordering samples $Q$ of size $|Q| = m$ such that any embedding in $\ell_p$ space in $d$ dimensions requires $d = \Omega(\sqrt{mt})$.*

4. *(k-NN) There exists a metric $\delta$ on $n$ points such that any embedding in $\ell_p$ space which preserves the k-NN ordering of $\delta$ must have $d = \Omega(k)$ dimensions.*

*When $p$ is a positive odd integer or when $p = \infty$, the lower bounds decrease by a logarithmic factor, that is the lower bounds are respectively $\Omega(\sqrt{m}/\log m)$, $\Omega(k/\log k)$, $\Omega(\sqrt{m}/\log m)$, and $\Omega(\sqrt{mt}/\log(mt))$.*

Note that in the above statements, $V$ can be arbitrarily large: in the proofs below, we can choose subsets of required size inducing all the samples.

*Proof.* We consider the case of positive even $p$. The cases of positive odd or infinite $p$ are analogous.

1. Choose an arbitrary set $V$ of size $\sqrt{m}$. By Theorem 40, there exists a non-contradictory sample set $Q$ of size $\Theta(m)$ on point set $V$ such that any embedding into $\ell_p$ space which is consistent with $Q$ must have dimension $\Omega(\sqrt{m})$.

2. Let $V$ be an arbitrary set of size $\sqrt{m} + (t - 1)$, and $V'$ be an arbitrary subset of $V$ of size $\sqrt{m}$. By the previous item, there exists a non-contradictory sample set $Q'$ of size $\Theta(m)$ on a set $V'$ that requires dimension $\Omega(\sqrt{m})$ dimensions. Let $V \setminus V' = \{v_1, \ldots, v_{t-1}\}$. For each $s' = (x, y^+, z^-) \in Q'$, define $s$ to be the $(t+1)$-tuple sample $s = (x, y^+, z^-, v_1^-, \ldots, v_{t-1}^-)$. Let $Q$ be the set of all such $(t+1)$-tuple samples.

   Next, we prove that $Q$ is non-contradictory. Since $Q'$ is non-contradictory, there is a metric $\delta'$ on $V'$ which is consistent with $Q$. Consider the following metric $\delta$ on $V$: for $x, y \in V'$, we set $\delta(x, y) = \delta'(x, y)$, and otherwise $\delta(x, y) = D$, where $D = 2\max_{x,y \in V'} \delta(x, y)$. It is easy to see $\delta$ satisfies triangle inequality, and is consistent with $Q$. Since every constraint in $Q'$ is implied by some constraint in $Q$, embedding preserving $Q$ must also preserve $Q'$, requiring $\Omega(\sqrt{m})$ dimension.

3. Choose an arbitrary set $V$ of size $\sqrt{mt}$. By the first item, there exists a non-contradictory sample set $Q'$ of size $O(mt)$ on a set $V$ that requires dimension $\Omega(\sqrt{mt})$. It suffices to show that there is a set of non-contradictory $Q$ of size $O(m)$ of $(t+1)$-tuple samples that imply all inequalities of $Q'$.

   Consider a metric $\delta$ on $V$ consistent with $Q'$. Denoting the $j$'th nearest neighbor of $x$ according to $\delta$ as $\pi_j(x)$, let

   $$Q = \cup_{x \in V} \{(x, \pi_1(x), \ldots, \pi_t(x)), (x, \pi_t(x), \ldots, \pi_{2t-1}(x)), \ldots\},$$

   where the adjacent samples share one item. We note that $Q$ is consistent with $\delta$, hence is non-contradictory. Finally, every inequality in $Q'$ is implied by the inequalities of $Q$: this is due to the fact that $\delta$ is consistent with $Q'$, and $Q$ implies all ordinal constraints of $\delta$ (as it implies the order of distances between each point and all its neighbors).

4. Choose an arbitrary set $V$ of size $k + 1$. By Theorem 40, there exists a non-contradictory sample set $Q$ on point set $V$ such that any embedding into $\ell_p$ space consistent with $Q$ must have dimension $\Omega(k)$. Consider a metric $\delta$ on $V$ consistent with $Q$. Since $|V| = k + 1$, k-NNs preserve all triplet comparisons of $\delta$, and therefore, any embedding of $V$ preserving the k-NN ordering has to be consistent with $Q$, hence requiring dimension $\Omega(k)$. □

# F   Other Results

In this section, we first extend our results to contrastive queries with more than two candidates. Then, we show that the problem of actually constructing the embedding consistent with given contrastive samples is NP-hard. Finally, we consider an *approximate* setting for contrastive learning, in which we only need to satisfy an $\alpha$-fraction of the constraints. We show that there exists an instance for which satisfying $\alpha \approx 0.77$ fraction of the constraints requires roughly the same number of dimensions as satisfying all constraints. On the other hand, we show that for $\alpha \le 1/2$, one dimension always suffices.

## F.1   Upper Bound for $t$-Negatives and $t$-Ordering Samples in $\ell_2$-norm

In this section, we consider two additional settings, in which each sample contains ordinal information about the distance between an anchor point and multiple (i.e. more than two other) points.

In the first setting ($t$-negatives), we are given a set $Q$ of $m$ samples, where each sample $s$ is a $(t+2)$-tuple $s = (x, y^+, z_1^-, \ldots, z_t^-)$. We say sample $s$ is satisfied by distance function $\delta$ if $\delta(x, y) > \delta(x, z_i)$ for all $1 \le i \le t$.

In the second setting ($t$-ordering), we are given a set $Q$ of $m$ samples, where each sample $s$ is a $t$-tuple $s = (x, y_1, \ldots, y_t)$, and we say sample $s$ is satisfied by distance function $\delta$ if $\delta(x, y_1) < \delta(x, y_2) < \cdots < \delta(x, y_t)$ for all $1 \le i \le t$.

**Theorem 44** ($t$-orderings, $t$-negatives)**.** *Let $Q$ be a set of $m$ non-contradictory $t$-ordering samples (resp. $t$-negative samples) on a set $V$. There is an embedding of $V$ into $\ell_2$-space $\mathbb{R}^{O(\sqrt{mt})}$ which is consistent with $Q$.*

*Proof.* For a set of $(t+2)$-tuple samples $Q$ on $V$ of size $m$, we define the *constraint graph* $G = (V, E)$ as follows: for each sample $(x_1, \ldots, x_{t+2}) \in Q$, we add $t+1$ edges $\{x_1, x_2\}, \ldots, \{x_1, x_{t+2}\}$ to $E$ (if they don't already exist).

First, we note that the constraint graph of $t$-orderings and $t$-negatives has arboricity $O(\sqrt{mt})$. Indeed, we add for each sample at most $O(t)$ edges to $G$, hence the total number of edges is at most $O(mt)$. By Fact 7, the arboricity of $G$ is $r = O(\sqrt{mt})$. By Theorem 9 there exists an embedding into $\ell_2$-space with dimension $r = O(\sqrt{mt})$ that satisfies the corresponding inequalities.                                   $\square$

## F.2   NP-Hardness for $d = 1$

In this section, we show that, empirical risk minimization for embedding into an $\ell_p$ space is NP-hard. Even in the realizable case and even for $d = 1$, finding an embedding satisfying constraints is NP-hard, by the reduction from the betweenness problem.

**Definition 45** (Betweenness)**.** *You are given a set of items $X$ of cardinality $n$ and a set of triplets $\{(a_1, b_1, c_1), \ldots, (a_m, b_m, c_m)\}$, such that $a_i, b_i, c_i \in X$ for all $i$. The goal of the betweenness problem is to find an order of items on $X$ so that for each $i$, $b_i$ is located between $a_i$ and $c_i$. That is, the goal is to find a bijection $r \colon X \to \{1, \ldots, n\}$ so that for each $i$ either $r(a_i) < r(b_i) < r(c_i)$ or $r(c_i) < r(b_i) < r(a_i)$ hold.*

[Opa79] shows that the decision version of the betweenness problem – i.e. checking whether such an ordering exists – is NP-hard.

**Theorem 46.** *Unless $P = NP$, there is no polynomial algorithm for finding an embedding into $\ell_2$ space for $d = 1$ in the realizable case.*

*Proof.* Let $A$ be an algorithm for finding an $\ell_2$ embedding for $d = 1$, which accepts the set of contrastive queries as an input. For contradiction, assume that in the realizable case the algorithm finds an embedding in time at most $T(n) = \text{poly}(n)$, where $n$ is the number of points.

Let $A'$ be the algorithm which executes $A$ for at most $T(n) = \text{poly}(n)$ iterations. This way, $A'$ runs on all inputs in time at most $T(n)$ and outputs an embedding satisfying the input constraints iff such an embedding exists.

We complete the proof by reduction from the betweenness problem. Let $\{(a_1, b_1, c_1), \ldots, (a_m, b_m, c_m)\}$ be the input for the betweenness problem. Then, we can represent constraint "$b_i$ is between $a_i$ and $c_i$" using two contrastive constraints $(a_i, b_i^+, c_i^-)$ and $(c_i, b_i^+, a_i^-)$. For example, if $r(b_i) < r(a_i) < r(c_i)$, then the constraint $(c_i, b_i^+, a_i^-)$ is violated; other cases are similar.

We execute $A'$ on this set of contrastive constraints. Since the algorithm finds a satisfying embedding iff such an embedding exists, we can check whether the contrastive constraints – and hence the original betweenness constraints – are satisfiable by checking the output of the algorithm. Hence, we can verify whether the set of betweenness constraints is satisfiable in the polynomial time, which contradicts NP-hardness of the problem and assumption that $P \neq NP$. $\qquad\square$

### F.3 Satisfying a Fraction of Constraints

In this section, we consider the settings when the embedding doesn't have to satisfy all the constraints. Instead, for some constant $\alpha$, we want to satisfy at least an $\alpha$-fraction constraints. We show the following separation in the $\ell_p$ case for any integer $p$.

**Theorem 47.** *For the embedding into $\ell_p$ space for $p \in \{1, 2, 3, \ldots\}$, the following hold.*

- *For any $\alpha \leq 1/2$, for any set of $m$ constraints, for any $d \geq 1$ there exists an embedding with dimension $d$ satisfying at least $\alpha m$ constraints.*

- *Let $\alpha^* \approx 0.77$ be the root of equation $H(x) = x$, where $H$ is the binary entropy function. Then for any $\alpha > \alpha^*$, there exists a set of $m$ non-contradictory constraints so that satisfying at least $\alpha m$ constraints requires dimension at least $\Omega(\sqrt{m})$ for even $p$ and at least $\Omega(\sqrt{m}/\log m)$ for odd $p$.*

**Notes** The theorem shows that for $\alpha \leq 1/2$, the problem trivializes, while for $\alpha > \alpha^*$, the problem is asymptotically as hard as in the case when we have to satisfy all constraints (up to $\log m$ factor for odd $p$). There is a gap between $1/2$ and $\alpha^* \approx 0.77$, and we hypothesize that $\alpha^*$ bound is the most likely one to be improved, due to the union bound used in the proof below.

*Proof.* The case $\alpha \leq 1/2$ follows by the probabilistic argument, using the observation that a random one-dimensional embedding satisfies half of the constraints in the expectation. It remains to handle the case $\alpha > \alpha^*$. For that, we construct a set of $m$ triplets, and, for a random labeling of $m$ triplets, we look at the induced labeling of each subset of $\alpha m$ triplets. For each individual subset, we will show the probability that its induced labeling is achievable is less than $1/\binom{m}{\alpha m}$. By the union bound, the probability that any of the induced labelings is achievable is less than $1$, implying that for at least one labeling, none of the induced labelings is achievable

$\ell_2$ **distance** We first consider the $\ell_2$-case, and below we describe how to handle $\ell_p$ distance for other integer $p$. By Lemma 41, there for any set $V$ of items, there exists a set $C$ of $m = \binom{n-1}{2}$ unlabeled triplets such that any its labeling is realizable. For a sufficiently large $n$, assume that $d < cn$ for some constant $c$ (depending on $\alpha$ and to be specified later). We will show that for $\alpha > \alpha^*$, there exists no subset of $C$ of size $\alpha m$ so that every its labeling is realizable by some embedding into a $d$-dimensional space. For that, we will use the following fact.

**Fact 48** ([War68]). *Let $m \geq t \geq 2$ be integers, and let $P_1, \ldots, P_m$ be real polynomials on $t$ variables of degree at most $s$. Let*

$$U(P_1, \ldots, P_m) = \{\mathbf{x} \in \mathbb{R}^t \mid P_i(\mathbf{x}) \neq 0 \text{ for all } i \in [m]\}$$

*be the set of points $\mathbf{x} \in \mathbb{R}^t$ which are non-zero in all polynomials. Then the number of connected components in $U(P_1, \ldots, P_m)$ is at most $(4esm/t)^t$.*

Similarly to [AAE$^+$24], we apply this fact to the following polynomials: for each triplet $(x, y, z)$, for a fixed embedding function $F$, we define a polynomial

$$P_{xyz} = \|F(x) - F(y)\|_2^2 - \|F(x) - F(z)\|_2^2 = \sum_{i=1}^{d}(F_i(x) - F_i(y))^2 - \sum_{i=1}^{d}(F_i(x) - F_i(z))^2$$

Denoting $V = \{x_1, \ldots, x_n\}$, all $P_{xyz}$ for $(x, y, z) \in C$ are polynomials over $nd$ variables $F_1(x_1), \ldots, F_d(x_1), \ldots, F_1(x_n), \ldots, F_d(x_n)$.

Importantly, when $(x, y^+, z^-)$ is satisfied by $F$, the polynomial is negative, while, when $(x, z^+, y^-)$ is satisfied by $F$, the polynomial is negative. Hence, different choices of labels of $C$ must correspond to the different sign combinations of polynomials. Fact 48 shows that the number of sign combinations of the polynomials – and hence the amount of possible labelings – is bounded by $(8em/nd)^{nd} \le (4en/d)^{nd}$, where we used $m = \binom{n-1}{2} < \frac{n^2}{2}$.

For any subset of $\alpha m$ constraints, there are $2^{\alpha m}$ possible induced labelings. On the other hand, as shown above, only $(4en/d)^{nd}$ of the labelings are achievable. Taking the ratio of these values, we get that the probability that an induced labeling is realizable is at most

$$\frac{(4en/d)^{nd}}{2^{\alpha m}} = 2^{nd \log_2(4en/d) - \alpha m}$$

As outlined above, since there are at most $\binom{m}{\alpha m}$ subset of $\alpha m$ constraints, we want this ratio to be at most $1/\binom{m}{\alpha m}$. By a well-known fact [TJ06], $\binom{m}{\alpha m} \le 2^{H(\alpha)m}$, where $H$ is a binary entropy function. Hence, the probability that any subset of $\alpha m$ induced constraints is satisfiable is at most

$$\frac{(4en/d)^{nd} \binom{m}{\alpha m}}{2^{\alpha (n-1)^2/2}} \le 2^{nd \log_2(4en/d) - \alpha m + H(\alpha)m} = 2^{m(H(\alpha) - \alpha + (nd/m) \log_2(4en/d))}$$

Since $m \ge (n-1)^2/2$, for a sufficiently large $n$ we have $nd/m < 3d/n$. Consider the case when $d < cn$ for some constant $c$. When $c < 4$, the last term $(3d/n) \log_2(4en/d)$ monotonically increases in $d$, and hence we have

$$H(\alpha) - \alpha + (nd/m) \log_2(4en/d) < H(\alpha) - \alpha + 3c \log_2(4e/c)$$

When $\alpha > \alpha^*$, where $\alpha^* \approx 0.77$ satisfies $\alpha^* = H(\alpha^*)$, we have $0 > H(\alpha) - \alpha$. Since $f(c) = 3c \log_2(4e/c)$ is continuous and strictly monotone for $c \in [0, 4]$ and $f(0) = 0$, there exists $c' > 0$ such that $H(\alpha) - \alpha + 3c' \log_2(4e/c') < 0$. Hence, when $d < c'n$, there exists a labeling of $m$ triplets, so that no subset of $\alpha m$ triplets is satisfiable.

$\ell_p$ **distances for positive integer** $p$    When $p$ is even, the above argument doesn't change. When $p$ is odd, we encounter the issue that

$$\|F(x) - F(y)\|_p^p - \|F(x) - F(z)\|_p^p = \sum_{i=1}^{d} |F_i(x) - F_i(y)|^p - \sum_{i=1}^{d} |F_i(x) - F_i(z)|^p$$

is not a polynomial. We address this issue similarly to [AAE+24]: for each coordinate $i$, we guess the order of points with respect to this coordinate. This introduces an additional factor of $(n!)^d = 2^{O(nd \log n)}$ in the number of possible sign combinations. The derivation is similar to the above, but we instead want the following inequality:

$$H(\alpha) - \alpha + (nd/m) \log_2(4en/d) + O((nd/m) \log n) < 0,$$

which holds when $d < cn/\log n$ for some constant $c$. $\qquad\square$

