# OpenReview forum: "Embedding Dimension of Contrastive Learning and $k$-Nearest Neighbors"
_NeurIPS.cc/2024/Conference — NeurIPS 2024 poster_

### Official Review · Reviewer_AWx5 · 2024-07-11

**Soundness:** 3
**Presentation:** 3
**Contribution:** 3
**Rating:** 6
**Confidence:** 3

**Summary:**

This paper establishes various asymptotic upper (and some lower) bounds on the dimensionality required so that there is an embedding satisfying various types of ordinal constraints on the pairwise distances. The constraints are either triplet constraints or k-nearest neighbor relations. The paper considers different $\ell_p$ embedding spaces for both types of constraints and derives bounds for each of them. The authors conduct experiments on CIFAR 10 / 100 that support their theoretically established bounds.

**Strengths:**

Originality
- I am not aware of any other work that tackles the question of required minimal dimensionality for ordinal distance constraints. This is despite both contrastive learning and k-nearest neighbor graphs being popular tools in machine larning. This makes the contribution of the paper relevant.

Quality
- While the paper is very formal, I think its numerous and general findings will be useful for many parts of the machine learning community, especially representation learning.

Clarity:
- The paper is very theoretical and most of it consists of proofs. This makes is a fairly challenging read. But overall, the authors do a very good job of providing high-level intuitive sketches of each proof before conducing it formally or select to showcase a weaker, but more intuitive version of a stronger statement proved in the appendix. This makes the writing clear.

Soundness:
- The proofs in the main paper seem sound to me, see questions below. I did not check the proofs in the appendix.

**Weaknesses:**

- *W1 Missing conclusion / future work section:* The paper ends abruptly after the experiments section. Please use some space to summarize your findings, point out potential limitations, and discuss how to develop this work further.

- *W2 Exact $d=\sqrt{m}$ in experiments:* The authors comment on the qualitatively different behavior if $d>\sqrt{m}$ or $d\leq \sqrt{m}$ in Fig 3. But they choose powers of 10 for m and powers of 2 for d, so that none of the runs is exactly at $d=\sqrt{m}$. It might make sense to choose the same base for both the exponential range of $m$ and that of $d$. Or to resolve the area around $d=\sqrt{m}$ more fine grained. The corresponding bound only states $d=O(\sqrt{m})$, which can include a constant. Why would one expect the behavior to change exactly at $d=\sqrt{m}$?

- *W3 Discussion of bounds for the approximate setting:* Table 1 and 2 do a great job at providing an overview of the results in the paper (and field). For context, it would be very useful to also state some bounds for an approximate version of the problem from related work, in which, e.g., only a share of $(1-\alpha)m$ of the constraints needs to be statisfied, or all constraints need to be satisfied up to an additive margin of $\alpha$. This relaxed setting should allow much lower dimensionalities. I am not sure if there are such results. If so, please discuss them. If not, this might go to the conclusion paragraph.


**Minor:**
- *W4 Description of contrastive learning:* Lines 34-37 seem to imply that all forms of contrastive learning operate on ordinal distance constraints of a fixed set of points. This is not quite true: For instance, in self-supervised constrastive learning the constraints are not over a fixed set but a potentially infinite set of all possible augmentations of all data samples. Moreover, triples of (anchor, positive and negative sample) are not always strictly understood as hard constraints on the distance. Since the negative sample is often chosen uniformly, it might even happen to be equal to the positive sample or the anchor (although only rarely). Often, the triplet is rather to be understood as the pair (anchor, positive) coming from some data distribution that is to be modeled and the pair (anchor, negative) coming from some noise distribution, see Gutmann and Hyvärinen (2010, 2012). This is not an issue for the paper at large, but does require some more rephrasing.

- *W5 Consistency between using big-O notation and not:* In several places, e.g., line 58 and footnote 1 the use of big-O notation is not very consistent. Line 58 speaks of $m\leq n^2$ while the accompanying footnote 1 only of $m = O(n^2)$, which does not preclude $m> n^2$ and only makes an asymptotic statement.

- *W6 Resolution of Figure 3:* is too low, especially for inspecting the small performance gaps between many of the runs.

- *W7 Experimental validation for the k-NN setting:* I appreciate that the paper is predominantly theoretical. But a brief experiment on the k-NN setting would be nice (and might only appear in the appendix). This could also serve as using an even more toy setting than the experiment in Fig 3: For instance, simply sample some random points, compute the k-NN graph and directly optimize the embedding in some embedding space to reproduce the same k-NN graph (instead of optimizing a parametric embedding as in Fig 3).

- *W8 Line 185:* Since the uniformly random choice in line 185 is the reason for the linear independence in Lemma 11 it would be nice to briefly mention this (without giving the proof) to build intuition about the proof sketch further.

**Typo:**
Line 269: "for any $[J]...$" --> "for any $j\in [J]...$"

**Questions:**

- *Q1:* I did not get the Chernoff bound arguments in line 236 and 240. Could you please elaborate?

- *Q2:* My understanding of the proof of Lemma 18 is that an embedding is constructed whose entries are indexed by $[L]$ (line 257), so that the embedding is of dimension $O(L) = O(r\log n)$ by Def 15. But the statement of Lemma 18 claims the square of this as embedding dimensionality. What do I miss?

- *Q3:* I understand the reason for line 264,265 to be that $|U(C_{j(y)}(x)| = O(log(n))$ (load property) is larger than $J=O(r)$ (line 256 and Fact 6). But why is $log(n) > r$?

- *Q4:* In line 270, we seem to need $J=r$ rather than only $J=O(r)$ (see Q3 and W5).

- *Q5:* Which basis of the logarithm is chosen in the proof of Cor 19? Since nothing is specified, I assume that it is the natural logarithm or to base 10. But since $m'$ is a power of 2, it seems it should be base 2 to make $log(m')$ an integer. Please clarify.

- *Q6:* I do not get the computation in the proof of Cor 19: $r \sum_{i=1}^{\log(m')} 2^i = r * (2^{\log(m')+1} -2) = 2r(m'-1) \neq rm'$.

- *Q7:* Line 294 speaks of $R=\Theta(k^2log(n))$ and $\alpha = \Theta(k^{-2})$. But since $p= O(k^{-1})$ (line 300), we have $\gamma = (1-p)p/k= O(k^{-2})$, $\alpha = O(k^{-3})$, and $R=O(k^3log(n))$. Should it perhaps be $\gamma = (1-p)p$?

- *Q8:* Please repeat what $w(x,y)$ means in line 317.

- *Q9:* I am a bit confused by the term "non-contradictory" in line 59 and Fact 51. In the triplet setting the distance constraint is phrased as $\leq$ not $<$, so the constant embedding always satisfies all constraints. Should these be strict inequalities?

**Limitations:**

Limitations are not explicitly discussed. Possible limitations might be the asymptotic nature of the statements (for which m, k, n do they start to hold)? What are the constants hidden in the big-O notation? Perhaps also the fact that many triplets are generated by sampling at least the negatives, which might let the total number of constraints considered in a real world contrastive learning setting be as high as $m=O(n^2)$, so that the bound $\sqrt{m}=n$ becomes very high.

---

> ### Author Rebuttal · Authors · 2024-08-07
>
> We thank the reviewer for your thoughtful, detailed, and precise comments. Please see our reply below:
>
> ### W1 Missing conclusion / future work section
>
> Thanks, please see the global response above for a proposed conclusion / future work section.
>
> ### W2 Exact $d=\\sqrt{m}$ in experiments
>
> Thank you! We first would like to clarify that the behavior changes around $\\sqrt{m}$ per our theory, not exactly at $\\sqrt{m}$. We changed the text to avoid confusion.
>
> We expanded experiments which accurately capture the regime around the $d=\\sqrt{m}$ threshold. In particular, we consider $m = c \\cdot d^2$ for $c=0.5, 0.75, 1, 1.5, 2, 2.5, 3$. The corresponding mean accuracies are $99\\%, 99\\%, 98\\%, 97\\%, 96\\%, 95\\%, 95\\%$.
>
> ### W3 Discussion of bounds for the approximate setting
>
> Thanks, we’ve added a discussion of the approximate setting, in which we give a full characterization of your suggested setting. It exhibits a surprising behavior: for $\\alpha \\ge 1/2$, one dimension always suffices, and for constant $\\alpha < 1/2$ it becomes as hard as the “exact” setting. We will be happy to outline it during the discussion period if you are interested.
>
> ### W4 Description of contrastive learning
>
> Thank you, we clarified that in the paper we consider one of the most common forms of contrastive learning.
>
> ### W5: Consistency between using big-O notation and not
>
> Thank you, we replaced $O(n^2)$ with $n^2$ in the footnote. Generally in this paper, we are interested in the asymptotic behavior.
>
> ### W6: Resolution of Figure 3
>
> Thank you, we added figures with higher resolutions.
>
> ### W7: Experimental validation for the k-NN setting
>
> We first would like to note that our construction for the upper bound for k-NN is theoretical - even for $k=1$ and mild $n$, the dimension significantly exceeds a trivial bound of $n$. In the general response, we show the experiments for k-NN using the setup similar to that from the paper, with the objective of reconstructing the original k-NN.
>
> ### W8: Adding a line of intuition for linear independence
>
> Thank you, we added the intuition that random perturbation is used to preserve independence.
>
> ### Limitations
>
> Thank you, we discuss the limitations in the conclusion (please refer to our general response).
>
> ### Constants in O-notation
>
> For $\\ell_2$ construction, the constants in O-notation are within a factor of 4. For the k-NN construction, the hidden constants are very large, which makes it mainly of theoretical interest. We specified in the conclusion that the k-NN construction can likely be improved.
>
> ### Quadratic number of constraints
>
> It’s indeed the case that the dimension might be as high as $\\Theta(n)$. But note that this is unavoidable in the worst case: using contrastive samples, one might recover full comparison information, and [CI24] show that, to preserve this information, $n/2$ dimensions are required in the worst case.
>
> ## Technical Questions
>
> > Q1. Chernoff bound in 236 and 240
>
> In 240, we have $L$ independent indicator random variables $A^j(x,y)$, each occurring w.p. $\\Omega(1/r)$. Denote $X = \\sum_j A^j(x,y)$. The expectation $E[X]$ is at least $E[X] = \\sum_j E[A^j(x,y)] = \\Omega(\\log{n})$. (since $L = \\Theta(r\\log{n})$).
> By Chernoff,
> $Pr[|X - E[X]| \\geq E[X]/2] < 2e^{-E[X]/12}$.
> In particular, $Pr[X = 0] < 2e^{-E[X]/12}$. Taking $L$ to be with sufficiently large constant, we get $Pr[X = 0] < 1/poly(n)$.
>
> In 236, Fix $x$. For $N^-(x) = \\{y_1,...,y_{O(r)}\\}$ we define $O(r)$ independent indicator variables $X_1,...,X_{O(r)}$, where $X_i = 1$ if $y_i$ and $x$ have the same color, and $X_i = 0$ otherwise. Denote $X = \\sum_i X_i$. Notice that $E[X] = O(1)$. By Chernoff, $Pr[X > c\\log{n}] \\leq Pr[X > (1+(c/E[X])\\log{n})E[X]] <  e^{-(c\\log{n}/2)} < 1/poly(n)$ (here $c$ is the constant of the load). Therefore, it holds for all $x$ w.h.p. by union bound.
>
> > Q2. Embedding dimension in Lemma 18
>
> In the NCC embedding, we have $L = \\Theta(r \\log{n})$ blocks, where each block is a standard unit vector of dimension $\\Theta(r\\log{n})$. Intuitively, the reason each unit vector is taken from this dimension is that we need $\\Theta(r\\log{n})$ distinct standard unit vectors.
>
> > Q3.
> > Do you require $|U(C_{j(y)}(x))| > J$?
>
> The only thing we require in those lines is that $|U(C_{j(y)}(x))| > 1$, because we only need to pick some vector distinct from that of $y$ (and since $j(y) \\in J$, we are guaranteed all other backwards neighbors choose a different vector for this block).
>
> > Q4,Q6,Q7.
> > In line 270, we seem to need $J=r$ rather than only $J=O(r)$. In Cor 19: $\\sum_{i=1}^{\\log{m'}} 2^i = 2(m'-1)$ and not $m'$. In Line 294, $\\gamma$ should be $\\gamma = (1-p)p$.
>
> Thank you for pointing out these issues. These are typos arised due to refactoring the proofs. We fixed them, which did not affect correctness or the final result.
> * Q4: In Lem 18, the exact distance should be $L - 1$ or $L$ (not $r-1$ or $r$ as written) - each $i \\in [L] \\setminus \\{j(y)\\}$ contributes $1$ to the distance (see lines 257-262 and 263-265).
> * Q6: we fixed the sum.
> * Q7: $\\gamma$ should be indeed $\\gamma = p(1-p)$. We also changed $\\gamma$ to $\\gamma/k$ in a few expressions due to this typo. This doesn't affect dependence on $k$ in the final bound.
>
> Q4, Q6, Q7 are now fixed.
>
> > Q5. Basis of the log in Cor 19
>
> By $\\log m'$ we denote the binary logarithm of $m'$ (i.e. $\\log_2$).
> We added the base of the logarithm in this place in all equations.
>
> > Q8. Repeating what $w(x,y)$ means in line 317
>
> We added a clarification that “$w(x,y)$ is the ranking of edge $\\{x, y\\}$ if the edges are sorted by the decreasing order of distances”
>
> > Q9. Should inequalities in line 59 and fact 51 be strict?
>
> Thank you! The inequalities between distances in the paper are strict. $\\le$ is a typo, which we fixed.
>
> Please let us know whether we addressed your concerns, and thank you again for your comprehensive review!

---

> ### Comment · Reviewer_AWx5 · 2024-08-08
>
> Thank you very much for your detailed response, which clarified many of my issues. I have a few remaining follow-up questions / comments.
>
> **W3:** This dependence on $\alpha \leq 0.5$ sounds intriguing! I'd be interested in a reference and suggest to add this for context to the background section of the paper.
>
> **W7:** Can you give a rough guidance on the (n,k) regime in which your kNN bound would be an improvement over the trivial bound of $n$? I.e. how large is the constant in the big-O notation? I appreciate the additional experiments on the $k$-NN graph. Perhaps a table showing the required dimension like you provide in the general comment for $\ell_2$ embeddings would be useful.
>
> **Q1 Chernoff bounds** I got the argument for line 240. Unfortunately, I am not very familiar with Chernoff bounds and for line 236, I only found the statement $P(X \geq (1+\delta)E(x))\leq e^{-\delta^2E(X)/(2+\delta)}$ online which looks similar to what you use, but is not quite the same. Could you please state the version of the Chernoff bound that you are using and a reference?
>
> **General comment on $\ell_2$ embeddings:** I am surprised that the required dimension decreases with $n$ for certain $m$-values. Could you please give some intuition why this happens?

---

> > ### Author Response · Authors · 2024-08-09
> > **Reply: W3**
> >
> > Thank you for the fast reply! We are happy to elaborate on these points. Please let us know if you have any questions remaining. If we addressed all your questions, we would be grateful if you could reconsider the score.
> >
> > ## W3: This dependence on $\\alpha \\le 0.5$ sounds intriguing! I'd be interested in a reference and suggest to add this for context to the background section of the paper.
> >
> > In the following, for simplicity, we changed the meaning of $\\alpha$ compared your review: we are interested in satisfying $\\alpha$-fraction of constraints, instead of $(1-\\alpha)$-fraction.
> > Here is what we can say quickly (we belive this can be improved with more work):
> >
> > * For any set of $m$ triplet constraints, a random one-dimensional $\\ell_p$ embedding satisfies at $m / 2$ constraints in expectation, hence there exists an embedding which satisfies at least $m/2$.
> > * Let $\\alpha^* \\approx 0.77$ be the root of equation $H(x) = x$, where $H$ is the binary entropy function. Let $\\alpha$ be a constant greater than $\\alpha^*$. Then for any $m$, there exists a set of $m$ constraints so that satisfying at least $\\alpha m$ constraints in $\\ell_2$ embedding requires dimension at least $\\Omega(\\sqrt{m})$.
> >
> > **Satisfying $\\alpha m$ constraints for constant** $\\alpha > \\alpha^*$
> >
> > We use the following fact (see Section 3 of [AAE+24] for more details):
> >
> > **Fact.** Consider a set of $m$ triplets $\\{(x_i, y_i, z_i)\\}_{i=1}^m$. Then the number of sets $T \\subseteq S$ such that there exists an embedding satisfying the following conditions is at most $(8 e m / (nd))^{nd}$.
> > * The embedding lies in the $d$-dimensional $\\ell_2$ space.
> > * For each triplet $(x_i, y_i, z_i)$ from $T$, the correct constraint according to the embedding is $(x_i, y_i^+, z_i^-)$.
> > * For each triplet $(x_i, y_i, z_i)$ from $S \\setminus T$, the correct constraint according to the embedding is $(x_i, z_i^+, y_i^-)$.
> >
> > In other words, for fixed $n$ and $d$, the number of different labelings (i.e. choices of triplets for which the first candidate is the closest to the anchor) *we can achieve* depends on $m$ polynomially, not exponentially.
> > This means that, when $m$ sufficiently large, the number of achievable labelings is less than $2^m$.
> > Hence, some of the labelings are not achievable, which is used by [AAE+24].
> >
> > Our idea is similar. This is the outline:
> > * For each subset of triplets of size $\\alpha m$, there are at most $(8 e \\alpha m / (nd))^{nd}$ possible labelings.
> > Meaning, if we choose a random labeling, the probability that this labeling is achievable is at most $(8 e m / (nd))^{nd} / 2^{\\alpha m}$.
> > We choose $m$ so that the probability is less than $1 / {m \\choose \\alpha m}$.
> > * Let's choose a random labeling of $m$ triplets, and look at the induced labeling of each subset of $\\alpha m$ triplets. For each individual subset, the probability that the labeling is achievable is less than $1 / {m \\choose \\alpha m}$. By the union bound, the probability that any of the induced labelings is achieavable is less than $1$.
> > * Hence, by the probabilistic argument, there exists at least one labeling of $m$ triplets so that none of the induced labelings of any subset of $\\alpha m$ triplets is achievable.
> > In other words, if we choose constraints according to this labeling, then for any embedding, there is no subset of $\\alpha m$ satisfied constraints.
> >
> > It remains to show that for $\\alpha > \\alpha^*$, there exists $m$ such that
> > $$
> > (8 e \\alpha m / (nd))^{nd} / 2^{\\alpha m} < 1 / {m \\choose \\alpha m} \\iff (8 e m / (nd))^{nd} < 2^{\\alpha m} \\cdot {m \\choose \\alpha m},
> > \qquad(*)
> > $$
> >
> > and for that we want the right-hand side to grow exponentially in $m$.
> > We use a well-known fact that
> > $${m \\choose \\alpha m} = 2^{m H(\\alpha)} \\cdot poly(...),$$
> >
> > where $H$ is the binary entropy function.
> > Since we want $2^{\\alpha m} \\cdot 2^{m H(\\alpha)}$ to grow exponentially in $m$, we need to guarantee that $\\alpha > H(\\alpha)$, which holds for $\\alpha > \\alpha^*$.
> > For $m = cnd / (\\alpha - H(\\alpha))$ for some constant $c$, we have
> > $$(8 em / (nd))^{nd} = (8ec \\alpha / (\\alpha - H(\\alpha)))^{nd}$$
> >
> > and
> > $$2^{\\alpha m} \\cdot {m \\choose \\alpha m} = 2^{cnd} \\cdot poly(...),$$
> >
> > and the desired inequality $(*)$ is satisfied for some constant $c$ (depending on $\\alpha$).
> >
> > Finally, similarly to the paper, by choosing $n = \\sqrt{m}$, we get $\\Omega(\\sqrt{m})$ on dimension.
> >
> >
> > **Notes.** There is a gap between $1/2$ and $\\alpha^* \\approx 0.77$.
> > Most likely, one can improve the $\\alpha^*$ bound since the union bound is too coarse.

---

> > ### Author Response · Authors · 2024-08-09
> > **Reply: Other Questions**
> >
> > > W7: Can you give a rough guidance on the $(n,k)$ regime in which your kNN bound would be an improvement over the trivial bound of $n$? I.e. how large is the constant in the big-O notation?
> > I appreciate the additional experiments on the $k$-NN graph. Perhaps a table showing the required dimension like you provide in the general comment for $\\ell_2$ embeddings would be useful.
> >
> > We note that the constants in the proof (including multiple 100's) were chosen for the sake of simplicity of the calculations, leading to very large total constant.
> > We believe that the constants can be decreased so that the final constant is approximately $10^4$. In this case, the final dependency would be $10^4 k^7 \\log^3 n$ which for very small $k$ might make it applicable for $n$ of order $10^8$-$10^9$. With some further work, one might be able to improve the constant event further but we believe that the more important (and closely related) question is improving the dependence on $k$.
> >
> > Overall, we would like to emphasize that the main contribution for the k-NN settings is theoretical, showing that the dependence on $n$ is at most poly-logarithmic.
> > This is very surprising, since, intuitively, k-NN still requires $n^2$ contrastive constraints: in particular, you need to compare the closest neighbor with all other neighbors.
> > In general, as our paper shows, $n^2$ constraints might require $\\Omega(n)$ dimension, but we show that these constraints have a sufficiently good structure to avoid linear dependence on $n$.
> >
> > We don't claim that the $k$-NN construction is practical as is, and it's very likely that the dependence on all parameters can be improved substantially, which we leave as a future work.
> > In particular, one of the main reason for the large dimension is due to randomized sampling scheme used to get Lemma 22.
> > It is likely that a simpler deterministic scheme exists (which potentially might not require $\\alpha$-designs, further simplifying the construction), which would probably give better dependency both on $k$ and constants.
> >
> >
> > > Q1 Chernoff bounds I got the argument for line 240. Unfortunately, I am not very familiar with Chernoff bounds and for line 236, I only found the statement $P[X \\geq (1+\\delta)E[X]]\\leq e^{-\\delta^2 E[X]/(2+\\delta)}$ online which looks similar to what you use, but is not quite the same. Could you please state the version of the Chernoff bound that you are using and a reference?
> >
> >
> > We'be been aiming to simplify the presentation, which in result omitted some details.
> > We use the standard Chernoff bound $P[X \\geq (1+\\delta)E[X]] \\leq e^{-\\delta^2 E[X]/(2+\\delta)}$, and below we show the complete derivation.
> >
> > We have
> > \\begin{align*}
> >     P[X > c \\ln n]
> >     &= P[X > \\frac{c}{E[X]} \\ln n \\cdot E[X]] \\\\
> >     &= P[X > (1 + (\\frac{c}{E[X]} \\ln n - 1)) \\cdot E[X]]
> > \\end{align*}
> >
> > Next, we use the referenced Chernoff bound with $\\delta = (\\frac{c}{E[X]} \\ln n - 1)$.
> > Note that for $n \\ge e$ and $c \\ge 3 E[X]$, we have $\\delta \\ge (\\frac{3 E[X]}{E[X]} \\ln e - 1) \\ge 2$ and hence $2 + \\delta \\le 2 \\delta$ and $\\delta^2 / (2 + \\delta) \\ge \\delta / 2$.
> > Finally, by the Chernoff bound, we have
> > \\begin{align*}
> >     P[X > c \\ln n]
> >     &\\le e^{-\\delta E[X] / 2} \\\\
> >     &\\le e^{-(\\frac{c}{E[X]} \\ln n - 1) E[X] / 2} \\\\
> >     &= n^{-c/2} \\cdot e^{E[X] / 2}
> > \\end{align*}
> >
> > Since $E[X] = O(1)$ (as a sum of $O(r)$ i.i.d. random variables with expectation $1/r$),
> > the error probability is $O(n^{-c/2})$, which can be made arbitrarily small by the appropriate choice of constant $c$.
> >
> >
> > > General comment on $\\ell_2$ embeddings: I am surprised that the required dimension decreases with $n$ for certain $m$-values. Could you please give some intuition why this happens?
> >
> > Great question! Recall that the dimension is $O(r)$, where $r$ is the arboricity.
> > Arboricity is the measure of density of the graph (see also figures in the global reply).
> > For a fixed $m$, with the increase of $n$, the density of the graph naturally decreases, which leads to smaller arboricity, and hence smaller dimension. We'll elaborate on this in the experiment.

---

> > > ### Comment · Reviewer_AWx5 · 2024-08-10
> > >
> > > Thanks for your reply!
> > >
> > > **W3:** Could you please comment on why you can choose $m$ (or $c$) to make (*) true while maintaining $m \leq n^2$, so that the triplets are not contradictory? I also do not get the last bit on choosing $n=\sqrt{m}$.
> > >
> > > **W7:** Thanks for stating the constant! I do not mind the result being largely theoretical, especially with the additional discussion on why the poly-logarithmic dependence is surprising. Nevertheless, I encourage you to state the constant and the $n$-regime in which your bound becomes practically useful in the revision.

---

> ### Author Response · Authors · 2024-08-12
> **More on W3 and W7**
>
> Thank you again for your comments!
>
> > W3. Could you please comment on why you can choose $m$ (or $c$) to make (*) true while maintaining $m \\le n^2$, so that the triplets are not contradictory? I also do not get the last bit on choosing $n = \\sqrt{m}$.
>
> Let us present the construction in a slightly different way, which hopefully answers both of your questions.
>
> **Construction.**
> Consider a set of $n$ points.
> On this set of $n$ points, according to Lemma 41, there exists a set of $m = {n - 1 \\choose 2} \\approx n^2 / 2$ triplets $C$, such that for every labeling of $C$ there exists a metric consistent with this labeling.
>
> We will show that, in general, $d = \\Omega(n)$ is required so that for any labeling of $C$ there exists an embedding into a $d$-dimensional $\\ell_2$-space.
> Since for this construction $n = \\Theta(\\sqrt{m})$, this implies that $d = \\Omega(\\sqrt{m})$ is required in general to embed a set of $m$ non-contradictory constraints.
>
> **Proof.**
> For the sake of contradiction, assume that there is a sufficiently large $n$ such that every labeling of the above set of triplets can be embedded using dimension $d < c n$ for some constant $c < 1$ (to be specified later).
> As mentioned in the previous reply, the amount of possible labelings consistent with an embedding of dimension $d$ is at most $(8 em / (nd))^{nd} \\approx (4e n / d)^{nd}$, where $\\approx$ ignores low-order terms in the exponent.
>
> As we showed in the previous reply, not all labelings are embeddable as long as the above expression is less than
> $$2^{\\alpha m} \\cdot {m \\choose \\alpha m} \\approx 2^{m(\\alpha - H(\\alpha))} \\approx 2^{n^2 (\\alpha - H(\\alpha)) / 2}$$
>
> When $d < cn$, we have
> $$(4e n / d)^{nd} < (4e / c)^{c n^2} = 2^{n^2 c \\log_2 (4e/c)}$$
>
> for a sufficiently large $n$.
>
> Then, if $c \\log_2 (4e/c) < (\\alpha - H(\\alpha)) / 2$ (such $c$ exists), we have that $(4e n / d)^{nd} < 2^{\\alpha m} \\cdot {m \\choose \\alpha m}$ for a sufficiently large $n$, implying that not all labelings of $C$ are embeddable.
> This proves that $d = \\Omega(n)$ is required.
>
> Since in this construction $n = \\Theta(\\sqrt{m})$, this means that $d = \\Omega(\\sqrt{m})$ is required in general.
>
> > W7. Thanks for stating the constant! I do not mind the result being largely theoretical, especially with the additional discussion on why the poly-logarithmic dependence is surprising. Nevertheless, I encourage you to state the constant and the $n$-regime in which your bound becomes practically useful in the revision.
>
> Thank you, we will specify the constants and add a short discussion about the constants and the practical regime at the conclusion of the paper.

---

> > ### Comment · Reviewer_AWx5 · 2024-08-12
> >
> > Thanks for clarifying the argument further!
> >
> > Overall, I am happy with the rebuttal and have raised my score to 6.

---

> > > ### Author Response · Authors · 2024-08-12
> > >
> > > Thank you for reconsidering the score and for your valuable comments!

---

### Official Review · Reviewer_wQgD · 2024-07-12

**Soundness:** 3
**Presentation:** 2
**Contribution:** 3
**Rating:** 5
**Confidence:** 1

**Summary:**

This paper discusses the number of compressed dimensions that satisfy the triplet or kNN constraints. In particular, theoretical results are derived for various distance measures as well as $L_2$.

**Strengths:**

The paper obtains intuitive and useful results. To this end, it introduces the concept of arboricity and obtains theoretical results.

**Weaknesses:**

Although this paper is a theory paper, I am strongly interested in whether the embedding is actually obtained. Let's say we have 1,000,000 vectors, where the dimensionality of each vector is 1,000. Our task is to preserve the top-500 ranking for each vector. Here, there are no encoders. We have vectors only. Can we practically set the valid $h$ beforehand (without knowing the final validated result)? To compute $F$, what computation do we really need? Is it fast or slow? The current experimental section does not contain the details of such a process.

**Questions:**

See te weaknesses section.

**Limitations:**

Not discussed.

---

> ### Author Response · Authors · 2024-08-12
> **Rebuttal**
>
> Thank you for your comments.
>
> > ... Whether the embedding is actually obtained ...
> > To compute $F$, what computation do we really need?
>
> Yes, all embedding constructions are explicit. The construction for $\\ell_2$ is in Section 2, and the construction for k-NN embeddings is in Section 3.
>
> > Here, there are no encoders.
>
> We would appreciate it if you could please clarify what you mean by encoders here.
>
> > Can we practically set the valid $h$ beforehand
>
>
> We are not exactly sure we understand this question. By $h$, we assume you mean the size parameter in the construction for $\\ell_2$. We don’t use this parameter in the k-NN construction, which your questions seems to be about.

---

### Official Review · Reviewer_ex1k · 2024-07-13

**Soundness:** 4
**Presentation:** 3
**Contribution:** 4
**Rating:** 8
**Confidence:** 4

**Summary:**

The paper "Embedding Dimension of Contrastive Learning and k-Nearest Neighbors" investigates the minimum embedding dimension required for representing datasets labeled with distance relationships in l_p-spaces, focusing on contrastive learning and k-Nearest Neighbor (k-NN) settings. The main findings suggest that the arboricity of associated graphs significantly impacts the design of these embeddings, providing tight bounds for the popular l_2-space and sufficient or necessary dimensions for other l_p-spaces based on the size of data (m for contrastive learning and k for k-NN).

**Strengths:**

One of the best papers that I read in a while! This paper is a standout piece, brilliantly blending strong theoretical insights with robust experimental validation. The authors have done an excellent job of presenting both lower and upper bounds, along with detailed experimental results, which really bring their work to life. It’s refreshing to see such a thorough demonstration in the field.

- Theoretical Depth and Rigor: The paper provides a rigorous mathematical framework for determining the embedding dimensions necessary for accurately representing data in different l_p-spaces.

- Practical Relevance: By exploring both contrastive learning and k-NN, the study addresses fundamental issues in machine learning that are highly relevant for practical applications, especially in areas like image recognition and nearest-neighbor classification.

- Comprehensive Analysis: The analysis spans multiple norms (l_2, l_inf, and general l_p) and provides both upper and lower bounds, tight bound, making the results robust and versatile for different scenarios.

**Weaknesses:**

Complexity of Proofs: Some of the proofs, especially those involving graph arboricity and its relation to embedding dimensions, are quite intricate and may not be easily understandable to readers without a deep background in theoretical computer science. Proofs might benefit from additional clarification or simplification for better understandabelity to non technical users. I had hard time reading and understanding them. (Disclaimer: I still need more time to fully digest them)

Empirical Validation: While experimental results are mentioned, the paper could strengthen its claims by expanding on these results, particularly how they compare with theoretical predictions across different settings and data scales.

Application for non-expert audiences: The paper presents significant theoretical results that could impact practical machine learning applications, but the presentation style and technical jargon might be inaccessible to non-specialists in theoretical computer science, which limits its potential audience and applicability.

The presentation could benefit from some polishing and simplification of the notation, though this is more a matter of preference than a true weakness.

**Questions:**

1. Bound Tightness (Section 1.1, Line 61): The paper discusses tight bounds for embedding in l_2 spaces for both contrastive learning and k-NN settings. Could you clarify the specific conditions or dataset characteristics that lead to these bounds being tight, and whether similar tightness is achievable in other l_p spaces?

2. Proof Complexity (Section 2, Lines 157-204): The proof of Theorem 8 relies on complex constructions involving vertex coloring and ordering by arboricity. Could you provide further intuition or simplified explanations to make this more accessible to readers without a deep background in graph theory?

2. Experimental Results (Section 4, Lines 328-347): You mention experiments on CIFAR-10 and CIFAR-100 with various embedding dimensions. Could you elaborate on how these experiments were structured, particularly how the embedding dimensions were selected and their impact on the model performance?

**Limitations:**

The main limitations that come to my mind are the limited empirical support and the complexity of the theoretical concepts.

---

> ### Author Rebuttal · Authors · 2024-08-07
>
> Thanks a lot for your positive feedback, thoughtful review and the suggestions.
>
> ### Complexity of Proofs
>
> Thanks a lot for pointing this out, we’ve substantially updated the exposition by adding illustrations of the key concepts (you can find some examples in the *global response* to all reviewers), as well as illustrations for some of the most complicated parts in Sections 2 and 3.
> We will also simplify the notation in the technical sections and add the intuitive explanations. To give an example, in Section 2, the arboricity $r$ implies that there exists an order of vertices so that each vertex has $O(r)$ preceding neighbors.
> We use it when constructing both parts of the embedding:
> * First, during construction of $F_1$, we want to control the inner products between the vertex and its preceding neighbors.
> This leads to at most $O(r)$ linear equations, and, by guaranteeing linear independence, it implies that $O(r)$ variables (that is, the embedding dimension of $O(r)$) is sufficient to guarantee the existence of the solution.
> * Second, during construction of $F_2$, for each vertex we update a single coordinate to normalize the vector. We want to avoid changes in the inner products between neighbors, and hence for each vertex we select the coordinate different from that of any preceding neighbor. Since there are at most $O(r)$ preceding neighbors, we require only $O(r)$ coordinates.
>
> ### Empirical Validation
>
> As per your suggestion, we added a major update to the experimental section by focusing on the various settings (you can find it in the *global response* above).
> In particular, the response includes:
> * $\ell_2$ construction based on Section 2;
> * k-NN experiments;
> * experiments on LLMs.
>
> Please note that in Figures 3a) and 3b), we consider different values of $m$, which corresponds to a wide range of data scales, having as much as $10^6$ samples (in the *global response*, we consider up to $10^7$ samples).
>
> ### Application for non-expert audiences
>
> We clarified the technical sections to make them more accessible.
> We have added a conclusion section (see a comment above) to summarize our main contribution, which we believe should be relevant to non-experts.
> In particular, our results provide guidance for the choice of embedding dimension in contrastive learning and k-NN applications, which we believe can be understood without diving into the theoretical details.
>
> ### Bound Tightness
>
> The bounds are likely to be tight when the set of constraints is dense, i.e. when there exists a subset of $\\Omega(\\sqrt{m})$ elements so that the samples are focused on these elements.
>
> For k-NNs, the lower bounds are likely to be tight in many cases, unless the dataset has some degeneracy.
> It is likely that the upper bounds for k-NNs are not tight, and the dependence on $k$ can be improved substantially.
>
> Please let us know if our response addresses your concerns.

---

> > ### Comment · Reviewer_ex1k · 2024-08-14
> > **Happy reviewer**
> >
> > Thanks for your hard work, your responses fully addressed my comments. Good luck!

---

> > > ### Author Response · Authors · 2024-08-14
> > > **Happy authors**
> > >
> > > Thanks a lot, very excited to hear that this is one of the best papers you've read in a while and your concerns have been fully addressed! :)

---

### Official Review · Reviewer_x73K · 2024-07-13

**Soundness:** 4
**Presentation:** 4
**Contribution:** 3
**Rating:** 6
**Confidence:** 4

**Summary:**

Paper studies the embedding dimension of contrastive learning and kNN problem.
In the first, we are given n points along with some constraints of the form (x,y, z1, z2, .., zm) which mean that x is closer to y and far from z1 z2  zm. Indeed, y is said to be positive label for x and z1, z2, zm are negative exmaples. It has been seen that such learning with + and - examples yield very good quality.

So the question studied in this work is if we were to embed the objects into vector space such that ||f(x) - f(y)|| < || f(x) - f(zi) || for all i, then what embedding dimension do we need?

Similarly, k-NN classifiers are used widely in ML. Here, we are given (x, y1 y2, .. yk) and want to preserve the ordering in the embedded space, i.e.,  ||f(x) - f(y_i)|| < ||f(x) - f(y_j)|| when i < j. In a generalization, we also require that y1.. yk are the kNNs in the embedded space also.

Main results:
Authors show for l2 metrics, we can embed into sqrt(m) dims for contrastive learning
They show slightly weaker results for l_infty metric and l_p metrics for general p.

More interestingly, for k-NN embeddings, they show embeddability into poly(k) dims for lp norm and k dims for l2 norm!

Main idea is to use arboricity as a key intermediate concept in showing embeddability. Indeed, a graph has arboricity r if it can be split as a union of r trees/ forests in general. Nice graph theoretic result states that graph with m edges has arboricity sqrt(m).
They then use this on the constraint graph resulting from the constraints placed by the embedding requirements (distances less than other distances). They show that if constraint graph has arboricity r, it can be embedded into l2 metric of dimension O(r). Moreover for kNN constraint graph they show that arboricity is O(k), so the result follows. Most relevant prior work [CI24] which gives some lower bounds.

They finally show some empirical results on CIFAR dataset showing that the obtained theoretical bounds can be met in practice.

**Strengths:**

It is very interesting to study embedding dimension. Embeddings are the cornerstone of future ML and low-dimensional embeddings representative of data is crucial. Contrastive learning and kNN methods are important tools, so preserving their structure is useful.

The paper has clean ideas using arboricity as intermediate concept. Paper is also well written.

**Weaknesses:**

The experimental section seems slightly less convincing. Indeed, the embeddings you use to validate are not the ones you show theoretical bounds for right? Or am I misunderstanding.

Can we show that such embeddings which preserve contrastive and kNN properties are as useful down the line as the original embeddings for various tasks? Some empirical study along these lines would be useful.

**Questions:**

I have asked my qns in the weakness section.

**Limitations:**

Paper does not have explicit limitations/ future work section and would benefit from one.

---

> ### Author Rebuttal · Authors · 2024-08-07
>
> Thanks a lot for your careful review. We’ve updated the experimental section to provide a more precise validation of the theoretical bounds (please the global response posted to all the reviewers above).
>
> Regarding the downstream applications of our embeddings, the answer to this question is two-fold:
> * k-NN embeddings themselves represent the main downstream task in their context. Such embeddings are used for recommendation systems, ranking, etc. Hence, preserving Top-k results in this context is exactly aligned with the objective of the downstream task.
> * The primary goal of our work is not to develop a scheme for computing embeddings better than the existing deep learning pipelines. It is our goal, however, to provide theoretical guidance for the choice of the embedding dimension in such architectures and do this in a theoretically rigorous fashion. Hence, we don’t expect the exact embedding constructions proposed in our theoretical bounds to perform better than the empirical embeddings which might be able to better capture the intrinsic structure of the data. It is only our goal to argue that the dimensionality used by these empirical constructions indeed suffices.
>
> Regarding conclusion/future work, please refer to the global response.
>
> Please let us know whether our response addresses your concerns.

---

### Author Rebuttal · Authors · 2024-08-07

## Additional Figures

Based on the reviews, we decided to add figures to clarify the proofs. Please see the attached PDF.

## Conclusion

As suggested by the reviewers, we will use the extra space of the final version to include the following discussion of the future work and to reiterate our findings:

> In this paper, we provide bounds on the necessary and sufficient dimension to represent a collection of contrastive constraints of the form “distance $(a,b)$ is smaller than distance $(a,c)$". This is a fundamental question in machine learning theory. In particular, it helps educate the choice of deep learning architectures by providing guidance for the size of the embedding layer. Our experiments illustrate the predictive power of our theoretical findings in the context of deep learning. We also believe that it gives rise to many interesting directions for future work depending on the exact desiderata (e.g. approximate versions, different choices of normed spaces, bicriteria algorithms, agnostic settings, etc.).

> While the distance comparison settings we consider play a central role in contrastive learning and nearest neighbor search, so far there has been no theoretical studies of their embedding dimension. Our work is the first to present a series of such upper and lower bounds in a variety of settings via a novel connection to the notion of arboricity from graph theory. As a follow-up, one can consider an improved embedding construction for k-NN: while the upped bound from Section 3 shows that dependence on $n$ is at most poly-logarithmic, the dependence on k can likely can be improved. Another interesting direction is tighter data-dependent bounds on dimension: while we provide fine-grained bounds in terms of arboricity - which are potentially much stronger than bounds in terms of the number of edges - they don’t necessary captures properties of dataset which can lead to sharper bounds.

## Additional Experiments

In this section, we present the experiments based on the reviewers' suggestions.  Please note that due to tight time constraints, the scale of the experiments below is limited. We will expand on these experiments in the camera-ready submissions.

### Our construction for $\\ell_2$-embeddings

In the table below, we show the dimension achieved by our construction from Section 2.
We consider these ranges of $n$ and $m$, since it allows $m$ to span from $m < n$ to $m > n^2$.

| | $m=10^2$ | $m=10^3$ | m=$10^4$ | m=$10^5$ | m=$10^6$ | m=$10^7$|
|---|---|---|---|---|---|---|
|$n=128$| 8 | 34 | 169 | 256 | 256 | 256 |
|$n=256$| 6 | 19 | 137 | 452 | 512 | 512 |
|$n=512$| 4 | 12 | 81 | 501 | 1024 | 1024 |
|$n=1024$| 4 | 8 | 43 | 362 | 1457 | 2048 |
|$n=2048$| 4 | 6 | 24 | 203 | 1488 | 3956 |

### Experiments for k-NNs

In the next table, we present results for k-NNs for $d=128$ and for various choices of $n$ and $k$.
We generate the k-NN using a ground-truth embeddings $e^*$.
For each element $u$, let $v_1^{(u)}(e^*), ..., v_{n-1}^{(u)}(e^*)$ be other elements, sorted by their distance to $u$ according to $e^*$.
For each $i \\in [k]$ and $j > i$, we generate contrastive samples $(u, v_i^{(u)}(e^*)^+, v_j^{(u)}(e^*)^-)$, and we train the neural network on this set of samples similarly to Section 4.
We denote the trained embeddings as $e$.

For each $n$ and $k$, we report the "accuracy" of preserving the k-NNs.
For each vertex $u \\in V$ and $i \\in [k]$, we compute the change of rank of the $i$'th neighbor w.r.t. $u$ in the new embeddings.
Formally, we find $j$ such that $v_{j}^{(u)}(e) = v_{i}^{(u)}(e^*)$, which contributes the loss $|i - j|$.
Finally, we define the accuracy as the average loss over all $u \\in V$  and $i \\in [k]$.

||k=1|k=2|k=4|k=8|k=16|k=32|
|---|---|---|---|---|---|---|
|n=10| 0.2 | 0 | 0.175 | 0.55 | - |-|
|n=100| 0.04 | 0.17 | 0.285 | 0.78 | 1.376 |2.316|
|n=1000| 0.007 | 0.367 | 0.882 | 1.5 | 2.2983|3.336|

The table shows that, as our theory predicts, dependence on $n$ is minor.

### Other experiments

1. We expanded experiments from Figure 3b) to accurately capture the regime around the $d=\\sqrt{m}$ threshold.
In particular, we consider $m = c \\cdot d^2$ for $c=0.5, 0.75, 1, 1.5, 2, 2.5, 3$.
The corresponding mean accuracies are $99\\%, 99\\%, 98\\%, 97\\%, 96\\%, 95\\%, 95\\%$.

2. We performed experiments similar to that from Figure 3b) on the Large Language Models tasks. We sampled $n=1000$ reviews from the Amazon sentiment dataset.
We get ground-truth embeddings using `paraphrase-MiniLM-L6-v2` model and train `stsb-roberta-base` model similarly to the settings from Figure 3.
We fix $d=128$ and consider $m = 10^3, 10^4, 10^5, 10^6$. The corresponding accuracies are $100\\%, 100\\%, 85\\%, 85\\%$, again showing the accuracy drop after $m$ significantly exceeds $d^2$.

---

### Decision · Program_Chairs · 2024-09-25

**Decision:**

Accept (poster)

**Comment:**

All reviewers support the acceptance of the paper with one reviewer especially enthusiastic.

Pros:

- study the important topic of embedding dimensions with relevant motivating applications of contrastive learning and k-nearest neighbor.

- comprehensive set of results for different norms (l2, l_p, l_infinity) and both upper and lower bounds

Cons:

- the experiments concerning the k-NN setting are lacking